# INKA, an integrative data analysis pipeline for phosphoproteomic inference of active kinases

Robin Beekhof[1,2,†] , Carolien van Alphen[1,2,†], Alex A Henneman[1,2,†] , Jaco C Knol[1,2],
Thang V Pham[1,2], Frank Rolfs[1,2], Mariette Labots[1], Evan Henneberry[2], Tessa YS Le Large[1,2],
Richard R de Haas[1,2], Sander R Piersma[1,2], Valentina Vurchio[3], Andrea Bertotti[3], Livio Trusolino[3],
Henk MW Verheul[1] & Connie R Jimenez[1,2,*]

## Abstract

Identifying hyperactive kinases in cancer is crucial for individualized treatment with specific inhibitors. Kinase activity can be discerned from global protein phosphorylation profiles obtained with mass spectrometry-based phosphoproteomics. A major challenge is to relate such profiles to specific hyperactive kinases fueling growth/progression of *individual* tumors. Hitherto, the focus has been on phosphorylation of either kinases or their substrates. Here, we combined label-free kinase-centric and substrate-centric information in an Integrative Inferred Kinase Activity (INKA) analysis. This multipronged, stringent analysis enables ranking of kinase activity and visualization of kinase–substrate networks in a single biological sample. To demonstrate utility, we analyzed (i) cancer cell lines with known oncogenes, (ii) cell lines in a differential setting (wild-type versus mutant, +/− drug), (iii) pre- and on-treatment tumor needle biopsies, (iv) cancer cell panel with available drug sensitivity data, and (v) patient-derived tumor xenografts with INKA-guided drug selection and testing. These analyses show superior performance of INKA over its components and substrate-based single-sample tool KARP, and underscore target potential of high-ranking kinases, encouraging further exploration of INKA's functional and clinical value.

**Keywords** cancer; computational tool; drug selection; kinase–substrate phosphorylation network; single-sample analysis
**Subject Categories** Cancer; Methods & Resources; Post-translational Modifications, Proteolysis & Proteomics
**Mol Syst Biol.** (2019) 15: e8250

## Introduction

Cancer is associated with aberrant kinase activity (Hanahan & Weinberg, 2011), and among recurrently altered genes, approximately 75 encode kinases that may "drive" tumorigenesis and/or progression (Vogelstein *et al*, 2013). In the last decade, multiple kinase-targeted drugs, including small-molecule inhibitors and antibodies, have been approved for clinical use in cancer treatment (Knight *et al*, 2010). However, even when selected on the basis of extensive genomic knowledge, only a subpopulation of patients experiences clinical benefit (Valabrega *et al*, 2007; Flaherty *et al*, 2010; Huang *et al*, 2014), while invariably resistance also develops in responders. Resistance can not only result from mutations in the targeted kinase or downstream pathways, but also from alterations in more distal pathways (Al-Lazikani *et al*, 2012; Trusolino & Bertotti, 2012; Ramos & Bentires-Alj, 2015). This complexity calls for tailored therapy based on detailed knowledge of the individual tumor's biology, including a comprehensive profile of hyperactive kinases. MS-based phosphoproteomics enables global protein phosphorylation profiling of cells and tissues (Jimenez & Verheul, 2014; Casado *et al*, 2016), but to arrive at a prioritized list of actionable (combinations of) active kinases, a dedicated analysis pipeline is required as the data are massive and complex. Importantly, a prime prerequisite for personalized treatment requires that the analysis is based on a single sample. This is pivotal in a clinical setting, where one wishes to prioritize actionable kinases for treatment selection for individual patients.

Different kinase ranking approaches have been described previously. Rikova *et al* (2007) sorted kinases on the basis of the sum of the spectral counts (an MS correlate of abundance) for all phosphopeptides attributed to a given kinase, and identified known and novel oncogenic kinases in lung cancer. This type of analysis can be performed in individual samples, but is limited by a focus on phosphorylation of the kinase itself, rather than the (usually extensive) set of its substrates. Instead, several substrate-centric approaches,

1 Medical Oncology, Cancer Center Amsterdam, Amsterdam UMC, Vrije Universiteit Amsterdam, Amsterdam, The Netherlands
2 OncoProteomics Laboratory, Cancer Center Amsterdam, Amsterdam UMC, Vrije Universiteit Amsterdam, Amsterdam, The Netherlands
3 Department of Oncology, Candiolo Cancer Institute IRCCS, University of Torino, Torino, Italy
*Corresponding author. Tel: +31 20 444 2340; E-mail: c.jimenez@vumc.nl
†These authors contributed equally to this work

focusing on phosphopeptides derived from kinase targets, also exist, including KSEA (Casado *et al*, 2013; Terfve *et al*, 2015; Wilkes *et al*, 2015), pCHIPS (Drake *et al*, 2016), and IKAP (Mischnik *et al*, 2016). The only single-sample implementation of substrate-centric kinase activity analysis is KARP and has been reported recently (Wilkes *et al*, 2017).

Neither a kinase-centric nor a substrate-centric phosphorylation analysis may suffice by itself to optimally single out major activated (driver) kinase(s) of cancer cells. To achieve an optimized ranking of inferred kinase activities based on MS-derived phosphoproteomics data for single samples, we propose a multipronged, rather than a singular approach. In this study, we devised a phosphoproteomics analysis tool for prioritizing active kinases in single samples, called Integrative Inferred Kinase Activity (INKA) scoring. The INKA algorithm combines direct observations on phosphokinases (either all kinase-derived phosphopeptides or activation loop peptides specifically), with observations on phosphoproteins that are known or predicted substrates for the pertinent kinase. To demonstrate its utility, we analyzed (i) cancer cell lines with known driver kinases in a single-sample manner, (ii) cell lines in a differential setting (wild-type versus mutant, +/− drug), (iii) pre- and on-treatment tumor needle biopsies from cancer patients, (iv) cancer cell panels with available drug sensitivity data, encouraging further exploration of INKA's functional and clinical value, and (v) colorectal cancer patient-derived xenograft (PDX) samples with INKA-guided drug selection. INKA code is available through a web server at www.INKAscore.org (updating) and as a zip file (Code EV1, current version). Data are available under PXD006616, PXD008032, PXD012565, and PXD009995.

## Results

### INKA: integration of kinase-centric and substrate-centric evidence to infer kinase activity from single-sample phosphoproteomics data

To infer kinase activity from phosphoproteomics data of single samples, we developed a multipronged data analysis approach. Figure 1 summarizes the data collection (Fig 1A) and analysis workflows (Fig 1B) of the current study. For in-house data generation, we utilized phosphotyrosine (pTyr)-based phosphoproteomics of cancer cell lines, patient-derived xenograft tumors, and tumor needle biopsies (Dataset EV2). Kinases covered by individual analysis approaches are detailed in Dataset EV3.

As a first component, phosphopeptides derived from established protein kinases (KinBase, http://kinase.com; Manning *et al*, 2002) are analyzed. Kinase hyperphosphorylation is commonly associated with increased kinase activity. This is the rationale for using the sum of spectral counts (the number of identified MS/MS spectra) for all phosphopeptides derived from a kinase as a proxy for its activity, and to rank kinases accordingly, as pioneered by Rikova *et al* (2007).

Second, kinase activation loop phosphorylation is analyzed. Although all kinase-derived phosphopeptides are already used in the first analysis above, here only phosphorylation of a kinase domain essential for kinase catalytic activity is considered for scoring, effectively doubling its contribution to the INKA score as a

weighing measure. Most kinases harbor an activation segment, residing between highly conserved Asp-Phe-Gly (DFG) and Ala-Pro-Glu (APE) motifs. Phosphorylation of residues in the activation loop counteracts the positive charge of a critical arginine in the catalytic loop, eliciting conformational changes and consequent kinase activation (Nolen *et al*, 2004). To identify phosphopeptides that are derived from a kinase activation segment, we use the Phomics toolbox (http://phomics.jensenlab.org; Munk *et al*, 2016). Subsequently, kinases are ranked after spectral count aggregation as described above.

Third, as a substrate-centric complement to the kinase-centric analyses above, and similar to a key ingredient in KSEA analysis (Casado *et al*, 2013), one can backtrack phosphorylation of substrates to responsible kinases as an indirect way to monitor kinase activity. Therefore, experimentally established kinase–substrate relationships listed by PhosphoSitePlus (Hornbeck *et al*, 2015) are used to link substrate-associated spectral counts to specific kinases, followed by kinase ranking.

Fourth, another substrate-centric analysis is included to complement the previous step. To date, databases logging experimental kinase–substrate relationships are far from complete, leaving a large proportion of phosphopeptides that cannot be mapped as a kinase substrate. Therefore, we apply the NetworKIN prediction algorithm (Linding *et al*, 2007; Horn *et al*, 2014) to observed phosphosites to generate a wider scope of kinase–substrate relationships. NetworKIN uses phosphorylation sequence motifs and protein–protein network (path length) information to predict and rank kinases that may be responsible for phosphorylation of specific substrate phosphosites. In our application, after applying score cutoffs to restrict the NetworKIN output to the most likely kinase–substrate pairs, kinases are ranked by the sum of all spectral counts associated with their predicted substrates.

Finally, we devised a method to integrate the four analyses as described above and to provide a single metric that can pinpoint active kinases in single biological samples analyzed by phosphoproteomics (Fig 1B, Materials and Methods). Specifically, for a given kinase, associated values in either of the two kinase-centric analyses are summed, and the same is done for the two substrate-centric analyses. Subsequently, the geometric mean of both sums is taken as an integrated inferred kinase activity, or INKA score. A non-zero INKA score requires both kinase-centric and substrate-centric evidence to be present. Furthermore, a skew parameter is calculated (0 for exclusively kinase-centric, 1 for exclusively substrate-centric, and 0.5 for equal contribution; see Materials and Methods), indicating to which extent the INKA score is derived from kinase-centric or from substrate-centric evidence, respectively. For kinases that are missing from PhosphoSitePlus and cannot be inferred by NetworKIN prediction, a separate kinase-centric ranking is performed to include these MS-observed enzymes in the analysis. This group involves 172 out of 538 established protein kinases considered in our analyses (Appendix Fig S1). For kinases inferred through PhosphoSitePlus/NetworKIN but not observed by MS, the reciprocal analysis is not performed, as kinases display overlapping substrate specificities precluding unequivocal assignment of a substrate to a specific kinase.

The results of an INKA analysis are visualized through different plots (Fig 1B). Individual analyses result in a bar graph with top 20 kinases. Integration by INKA scoring results in a scatter plot for all

kinases with an INKA score of at least 10% of that of the top-scoring kinase (with the horizontal position indicating the skew toward kinase-centric or substrate-centric evidence). For the top 20 kinases

(by INKA score), a ranked bar graph and a network of all inferred kinase–substrate connections are visualized as well (Fig 1B). The INKA analysis pipeline is available as a web service at

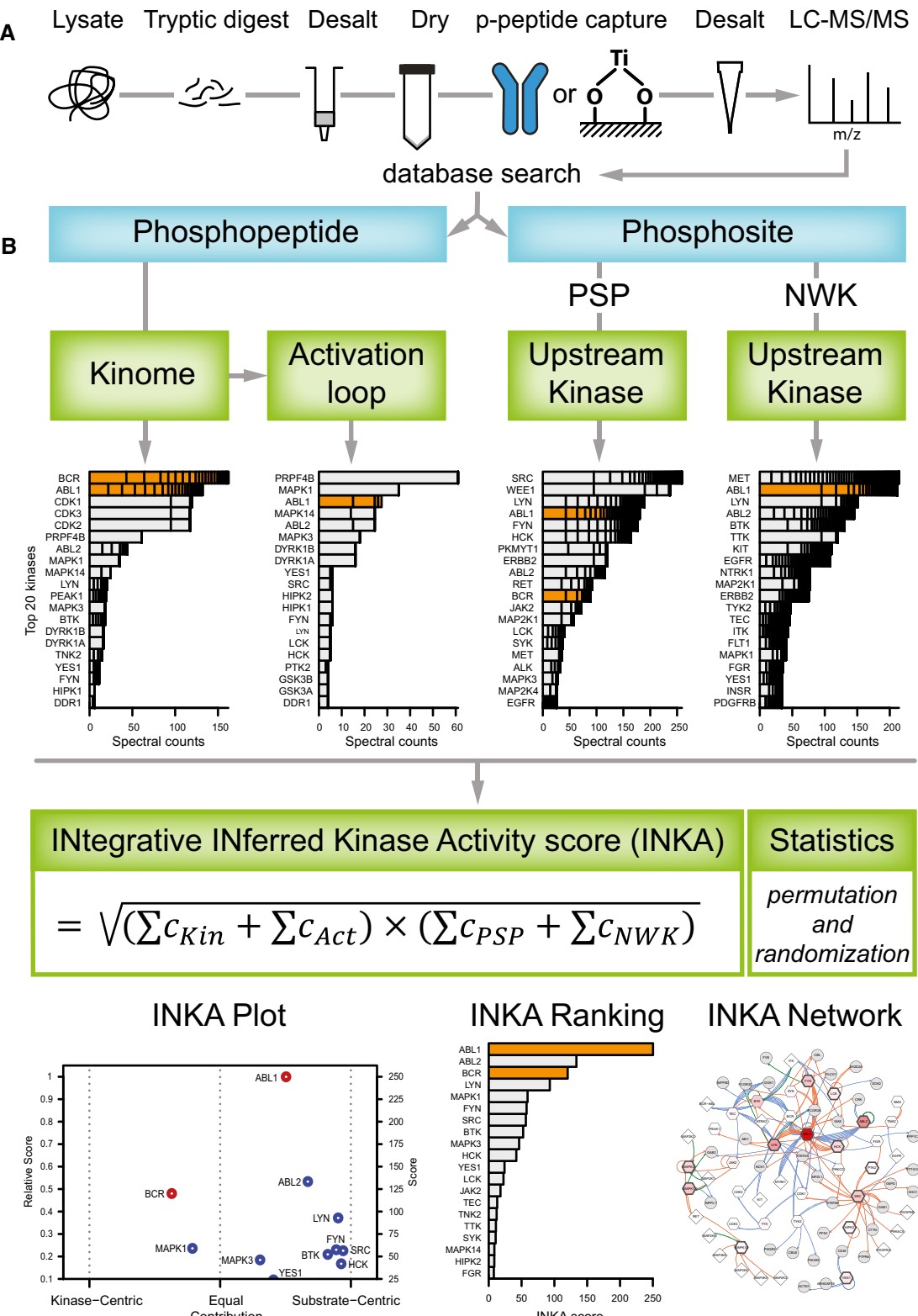

**Figure 1.**

◀

**Figure 1. Generic phosphoproteomics workflow and data analysis strategy.**

A Overview of an MS-based phosphoproteomics experiment. Proteins from a biological sample are digested with trypsin, and phosphopeptides are enriched for analysis by (orbitrap-based) LC-MS/MS. Phosphopeptides can be captured with various affinity resins; here, data were analyzed of phosphopeptides enriched with anti-phosphotyrosine antibodies and TiOx. Database-based phosphopeptide identification, and phosphosite localization and quantification are performed using a tool like MaxQuant.

B Scheme of INKA analysis for identification of active kinases in a single biological sample. Quantitative phosphodata for established kinases are taken as direct (kinase-centric) evidence, using either all phosphopeptides attributed to a given kinase ("kinome") or only those from the kinase activation loop segment ("activation loop"). Phosphosites are filtered for class I phosphosites (localization probability > 75%; Olsen *et al*, 2006), coupled to phosphopeptide spectral count data, and used for substrate-centric inference of kinases on the basis of kinase–substrate relationships that are either experimentally observed (provided by PhosphoSitePlus, "PSP") or predicted by an algorithm using sequence motif and protein–protein network information (NetworKIN, "NWK"). All evidence lines are integrated in a kinase-specific INKA score using the geometric mean of combined spectral count data ("C") for kinase-centric and substrate-centric modalities. Results are visualized in a scatter plot of INKA scores for kinases scoring ≥ 10% of the maximum ("INKA Plot"; horizontal shifts from the middle indicate evidence being more kinase-centric or more substrate-centric). For top 20 INKA-scoring kinases, a score bar graph ("INKA Ranking"), and a kinase–substrate relation network for pertinent kinases and their observed substrates ("INKA Network") are also produced.

http://www.inkascore.org, where the latest updated version is maintained and can be downloaded, while the current code is provided here as a zip file (Code EV1).

## INKA analysis of oncogene-driven cancer cell lines

To assess performance of the INKA approach, pTyr IP-based phosphoproteomic data were generated and analyzed for four well-studied cell lines with known oncogenic driver kinases: K562 chronic myeloid leukemia (CML) cells (*BCR-ABL* fusion), SK-Mel-28 melanoma cells (mutant *BRAF*), HCC827-ER3 lung carcinoma cells (mutant *EGFR*), and H2228 lung carcinoma cells (*EML4-ALK* fusion).

Figure 2 displays, per cell line, a row of bar graphs with the top 20 kinases for each of the four basic analyses (kinome, activation loop, PhosphoSitePlus, and NetworKIN) as well as the combined score analysis (INKA). Bars for known driver kinases are highlighted by coloring except for SK-Mel-28. For the latter cell line, driven by the serine/threonine kinase BRAF (not detected by pTyr-based phosphoproteomics), downstream driver targets in the MEK-ERK pathway (MAP2K1, MAP2K2, MAPK1, MAPK3) are highlighted (Fig 2B). The underlying data can be found in Dataset EV4. In general, drivers are among the top ranks of the four analysis arms albeit to somewhat different extents. Clearly, "kinome" analysis (Fig 2, first column of bar graphs) strongly suggests identification of hyperactive kinases, as was found previously (Rikova *et al*, 2007; Guo *et al*, 2008). However, the additional substrate-centric analyses provide more confidence that kinase phosphorylation correlates with target phosphorylation (i.e., kinase activity). This is reflected in top-ranking integrative INKA scores for all drivers (or a proxy in the special case of SK-Mel-28). Figure 3 shows scatter plots of INKA scores as a function of kinase-centric versus substrate-centric evidence contribution as well as inferred kinase–substrate relation networks for the top 20 kinases in the four cell lines. Larger versions of the networks can be found in the Appendix Figs S2–S5. Altogether, these results show that amplification-driven oncogenic kinases or constitutively active kinase(-fusions) rank high by INKA, in line with previous findings (Rikova *et al*, 2007; Guo *et al*, 2008).

In order to explore statistical significance, we permuted both experimental data (phosphopeptide–spectral count links for a sample) and annotation data (kinase–substrate links in complete PhosphoSitePlus and NetworKIN databases), and obtained INKA score null distributions to derive *P*-values for actually observed INKA scores. Almost all top 20 INKA scores in Fig 2 are significant. Higher INKA scores clearly correlate with lower *P*-values (Appendix Fig S6).

The INKA score was compared to KARP (Wilkes *et al*, 2017), another kinase activity ranking tool that can be used on single samples. KARP kinase activity ranking is based on substrate phosphorylation analysis in combination with kinase–substrate relations. For the four oncogene-driven cell lines, INKA outperformed KARP in assigning high ranks to the known drivers (Appendix Fig S7).

To further assess to what extent hyperactive kinases identified by the above INKA analysis represent actionable drug targets, we investigated public cell line drug sensitivity data from the "Genomics of Drug Sensitivity in Cancer" resource (Yang *et al*, 2013; GDSC, http://www.cancerrxgene.org, Dataset EV5). First, for K562 CML cells, INKA analysis pinpointed ABL1 as a prime candidate and inferred phosphorylation of its downstream signaling partners such as SRC-family kinases and MAPK1/3 (Figs 2A and 3A, Appendix Fig S2). Indeed, GDSC data indicate K562 to be sensitive to various ABL inhibitors. Second, INKA analysis of SK-Mel-28 (Figs 2B and 3B, Appendix Fig S3) revealed high-ranking activity of BRAF pathway partner MAPK3, SRC (a central node in the inferred kinase–substrate network) and PTK2, followed by other SRC-family members and BRAF pathway partner MAPK1. GDSC data indicate that SK-Mel-28 is sensitive to inhibition of BRAF and downstream MAP2K1/2 (MEK1/2) and, to a lesser extent, MAPK3/1 (ERK1/2). Based on the INKA data, one could test "combination therapy" with a PTK2 or SRC inhibitor (both not very effective as single agent) and a BRAF or MEK1/2 inhibitor. Interestingly SRC has been implicated in BRAF inhibitor resistance in *BRAF*-mutant melanoma cells and patient-derived tissues (Girotti *et al*, 2013). Third, for HCC827-ER3, a subline of HCC827 expressing mutant EGFR[E746-A750] but exhibiting *in vivo* acquired resistance to the EGFR inhibitor erlotinib (Zhang *et al*, 2012), INKA inferred high activity of EGFR and MET (Figs 2C and 3C). The latter are also highly connected nodes in the kinase–substrate network (Appendix Fig S4). Both EGFR and MET are known to be already highly active in the parental HCC827 line (Zhang *et al*, 2012; van der Mijn *et al*, 2016; Dataset EV4, Appendix Fig S8), which is relatively insensitive to MET inhibitors (GDSC). A kinase that could be involved in HCC827-ER3 erlotinib resistance is AXL. There are no PhosphoSitePlus-/NetworKIN-based substrate prediction data for AXL, precluding it getting an INKA score, but based on the kinase-centric part of the HCC827-ER3 analysis, it ranks second with relatively high counts (Figs 2C and 3C left panel), whereas it was not identified in parental HCC827 cells

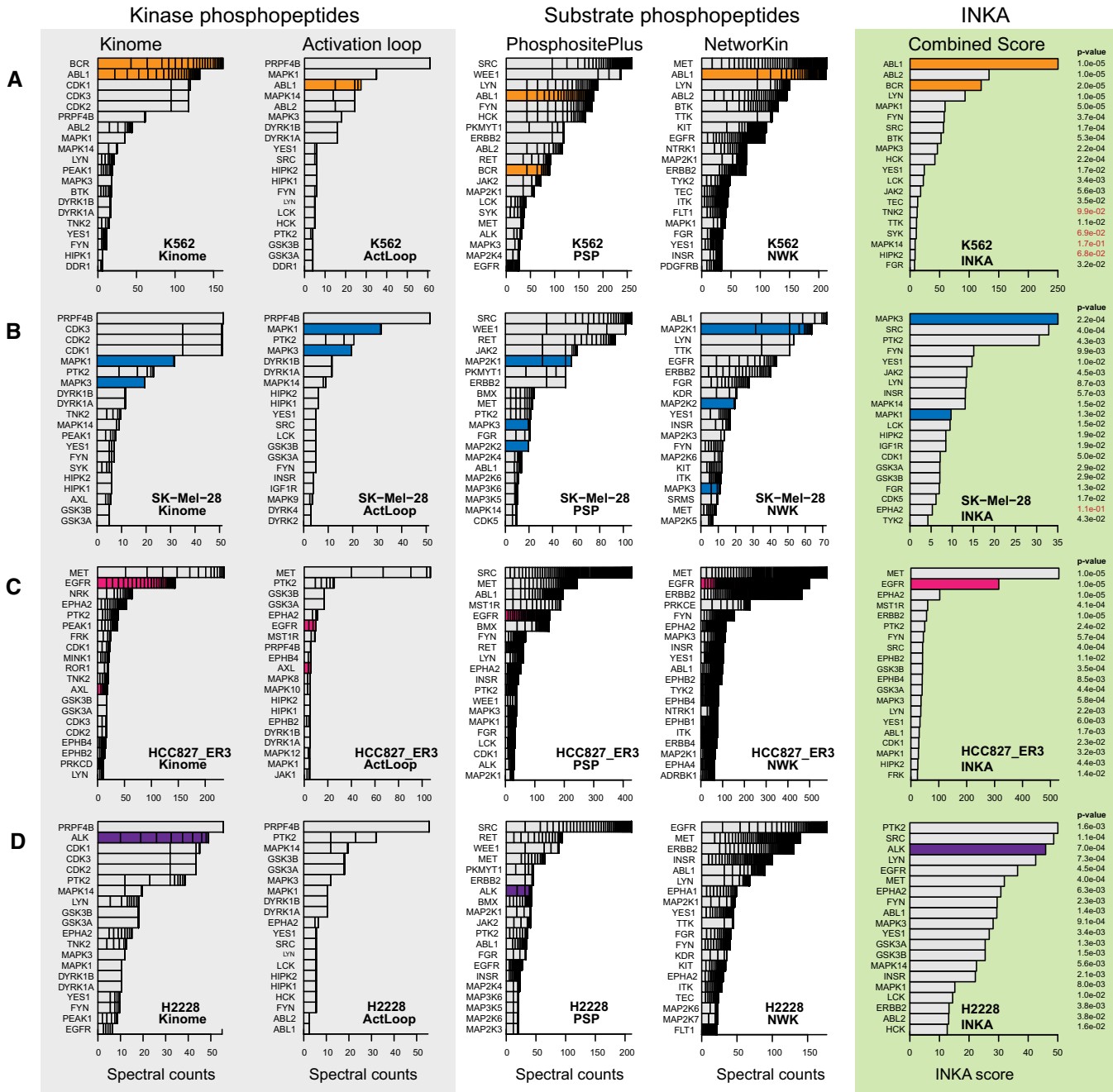

**Figure 2. Ranking of top 20 kinases in four cell line use cases by each of four lines of evidence and integrative INKA scoring.**

A  K562 chronic myelogenous leukemia cells with a *BCR-ABL* fusion. INKA score ranking indicates that ABL1/BCR-ABL (orange bars) exhibits principal kinase activity in this cell line, in line with a role as an oncogenic driver.

B  SK-Mel-28 melanoma cells with mutant *BRAF*. In the "kinome" analysis, CDK1, CDK2, and CDK3 share a second place, based on phosphopeptides that cannot be unequivocally assigned to either of them. INKA scoring implicates MAPK3 as the number one activated kinase. As SK-Mel-28 is driven by BRAF, a serine/threonine kinase that is missed by pTyr-based phosphoproteomics, downstream targets in the MEK-ERK pathway are highlighted by blue coloring.

C  Erlotinib-resistant HCC827-ER3 NSCLC cells with mutant *EGFR*. INKA scores reveal the driver EGFR (pink coloring) as second-highest ranking and MET as highest ranking kinase, respectively.

D  H2228 NSCLC cells with an *EML4-ALK* fusion. The driver ALK (purple coloring) is ranked as a top 3 kinase by INKA score, slightly below PTK2 and SRC.

Data information: For each cell line, bar graphs depict kinase ranking based on kinase-centric analyses (panel "Kinase phosphopeptides"), substrate-centric analyses (panel "Substrate phosphopeptides"), and combined scores (panel "INKA"). Bar segments represent the number and contribution of individual phosphopeptides (kinase-centric analyses) or phosphosites (substrate-centric analyses). Since substrate-centric inference attributes data from multiple, possibly numerous, substrate phosphosites to a single kinase, bar segments coalesce into a black stack in more extreme cases. *P*-values flanking INKA score bars were derived through a randomization procedure with $10^5$ permutations of both peptide-spectral count links and kinase–substrate links. *P*-values in red are above a significance threshold of *P* < 0.05.

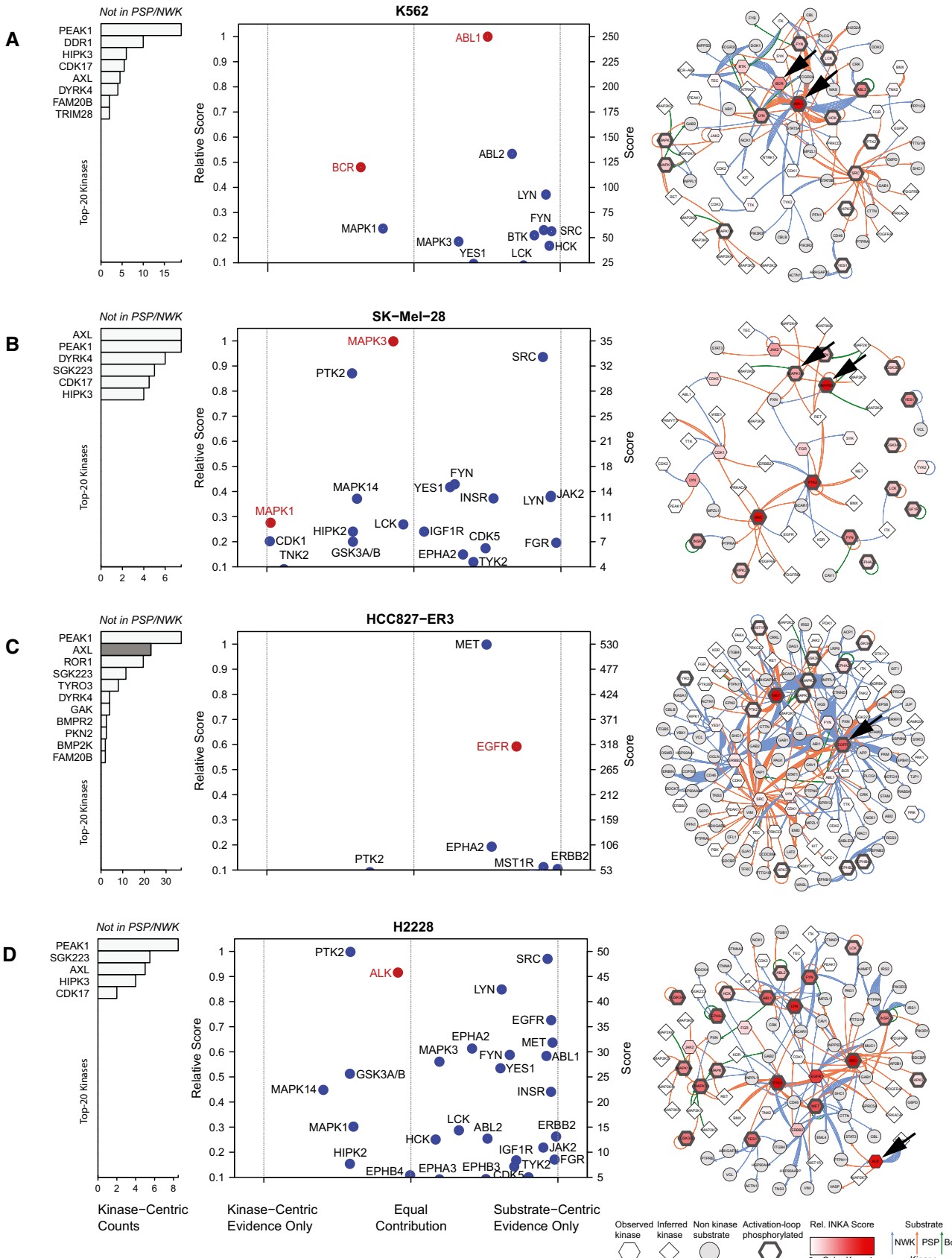

**Figure 3.**

**Figure 3. INKA plots and kinase–substrate relation networks for four oncogene-driven cell lines.**

A   K562 CML cells with a *BCR-ABL* fusion. ABL1 is the most activated kinase, with relatively equal contributions from both analysis arms. It is a highly connected, central node in the network.

B   SK-Mel-28 melanoma cells with mutant *BRAF*. Downstream MEK-ERK pathway members are highlighted in lieu of BRAF which is missed by the current pTyr-based workflow. MAPK3 is the top activated kinase. The network includes two clusters with highly connected activated kinases, MAPK1/3 and SRC, respectively.

C   Erlotinib-resistant HCC827-ER3 NSCLC cells with mutant *EGFR*. EGFR and MET are the most active, highly connected kinases. AXL, inactive in parental cells (see Appendix Fig S8), but associated with erlotinib resistance in this subline, can only be analyzed through the kinase-centric arm (pink bar highlighting).

D   H2228 NSCLC cells with an *EML4-ALK* fusion. ALK is a high-ranking kinase with roughly equal evidence from both analysis arms. Multiple highly active and connected nodes imply relative insensitivity to ALK inhibition, in line with previous functional data. Larger networks are shown in Appendix Figs S2–S5.

Data information: In INKA plots proper, the vertical position of kinases (drivers in red) is determined by their INKA score, whereas the horizontal position is determined by the (im)balance of evidence from kinase-centric and substrate-inferred arms of the analysis. Kinases not covered by PhosphoSitePlus (PSP) and NetworKIN (NWK) are visualized in a flanking bar graph.

(Appendix Fig S8). Indeed, AXL is a resistance hub in HCC827-ER3 (Guo *et al*, 2008; van der Mijn *et al*, 2016).

Fourth, for H2228 (Fig 2D), GDSC data indicate that effective inhibitors are rare, even when targeting its mutated oncogenic kinase, ALK (alectinib: IC50 4.4 μM). In the kinase–substrate network inferred for H2228 (Fig 3D, Appendix Fig S5), in addition to ALK, there are multiple hyperactive, highly connected kinases (e.g., PTK2, SRC, EGFR) as candidates for combination treatment even though single-agent treatment is ineffective. Where PTK2 and SRC are inhibited by several ALK inhibitors at IC50 concentrations (Davis *et al*, 2011), ruling these kinases out as co-targets, EGFR is implicated in reduced sensitivity to ALK inhibition in H2228, while dual inhibition of ALK and EGFR results in highly increased apoptosis (Voena *et al*, 2013). Furthermore, targeting both EGFR and ALK seems to have a synergistic effect (Li *et al*, 2016). In summary, the analysis of these cell lines encourages the idea that INKA may help prioritize therapeutic drugs.

**Testing the INKA approach with literature data**

To further test our strategy for prioritizing active kinases, we also examined phosphoproteome data of oncogene-driven cell lines from the literature (Guo *et al*, 2008; Bai *et al*, 2012; Fig 4, Dataset EV4). INKA analysis of data on *EGFR*-mutant NSCLC cell line H3255 (Guo *et al*, 2008) uncovered major EGFR activity in these cells, with EGFR ranking first, followed by MET (Fig 4A). In another study, the rhabdomyosarcoma-derived cell line A204 was associated with PDGFRα signaling (Bai *et al*, 2012), and INKA scoring of the underlying data accordingly ranks PDGFRα in second place (Fig 4B). In the same study, osteosarcoma-derived MNNG/HOS cells were shown to be dependent on MET signaling and sensitive to MET inhibitors (Bai *et al*, 2012). In line with this, INKA analysis clearly pinpointed MET as the major driver candidate in this cell line (Fig 4C).

Altogether, the above analyses of public datasets illustrated the capacity of INKA scoring to identify kinases that are relevant oncogenic drivers in diverse cancer cell lines at baseline.

**Testing the INKA approach in differential settings**

To explore the discriminative power of INKA scoring, we analyzed pTyr-phosphoproteomic data from wild-type versus mutant cells, and from untreated versus drug-treated cells as genetic and pharmacological dichotomies (Fig 5, Dataset EV4). First, we reanalyzed data from our laboratory (van der Mijn *et al*, 2014) to compare wild-type U87 glioblastoma cells with isogenic U87-EGFRvIII cells

overexpressing a constitutively active EGFR mutant. The EGFR INKA score was significantly higher and dominating in U87-EGFR-vIII relative to wild-type U87 (Fig 5A). Some other kinases also exhibited higher-ranking INKA scores, including MET and EPHA2, for which enhanced phosphorylation in U87-EGFRvIII has been previously documented (Huang *et al*, 2007; Stommel *et al*, 2007), as well as SRC-family members. In a treatment setting, INKA analysis clearly revealed a specific drug effect after targeting EGFR in U87-EGFRvIII, with the high, first-rank INKA score for EGFR at baseline being halved after treatment with erlotinib (Fig 5B).

Second, in a more clinical application, INKA scoring was applied to pTyr-phosphoproteomic data on patient tumor biopsies (Fig 5C and D). Biopsies were collected both before and after 2 weeks of erlotinib treatment to study intra-tumor drug concentrations within the framework of a phase I clinical study (standard dose, trial NCT01636908; Labots *et al*, in preparation). Patients were not assigned to erlotinib treatment based on molecular profiling. Nonetheless, the on-treatment biopsy from a patient with advanced head and neck squamous cell carcinoma showed a reduced INKA score and rank for EGFR as well as cell cycle-associated kinases (Fig 5C). Interestingly, in a pancreatic cancer patient, no residual EGFR activity could be inferred by INKA in a tumor biopsy after erlotinib treatment (Fig 5D). The limited patient material was available precluded replicate analysis, so results reported here are preliminary.

In Appendix Fig S9, differential analyses at the individual INKA component levels (kinome, activation loop, PSP, NWK) can be found. For the comparisons of U87 wild-type versus EGFR-mutant cells (Appendix Fig S9A) and untreated mutant cells versus erlotinib-treated mutant cells (Appendix Fig S9B), INKA components are similar. All four components indicate lower EGFR phosphorylation in wild-type relative to mutant cells and erlotinib-treated relative to untreated cells, respectively, while, e.g., MET phosphorylation is not affected by erlotinib treatment. For low-level input samples from patients (Appendix Fig S9C and D), the combined INKA score shows the more robust EGFR response, compared to the individual INKA components. Taken together, combination of the four INKA components into a single score averages out noise and results in a more robust kinase activity ranking than each individual component by itself, and can also be applied in a differential setting.

Third, to assess performance of INKA analysis in a preclinical model with clear-cut biological effects, we analyzed published phosphoproteome data (Bensimon *et al*, 2010) of human G361 melanoma cells after induction of genotoxic stress with neocarzinostatin, a radiomimetic that induces double-strand breaks (Fig 5E

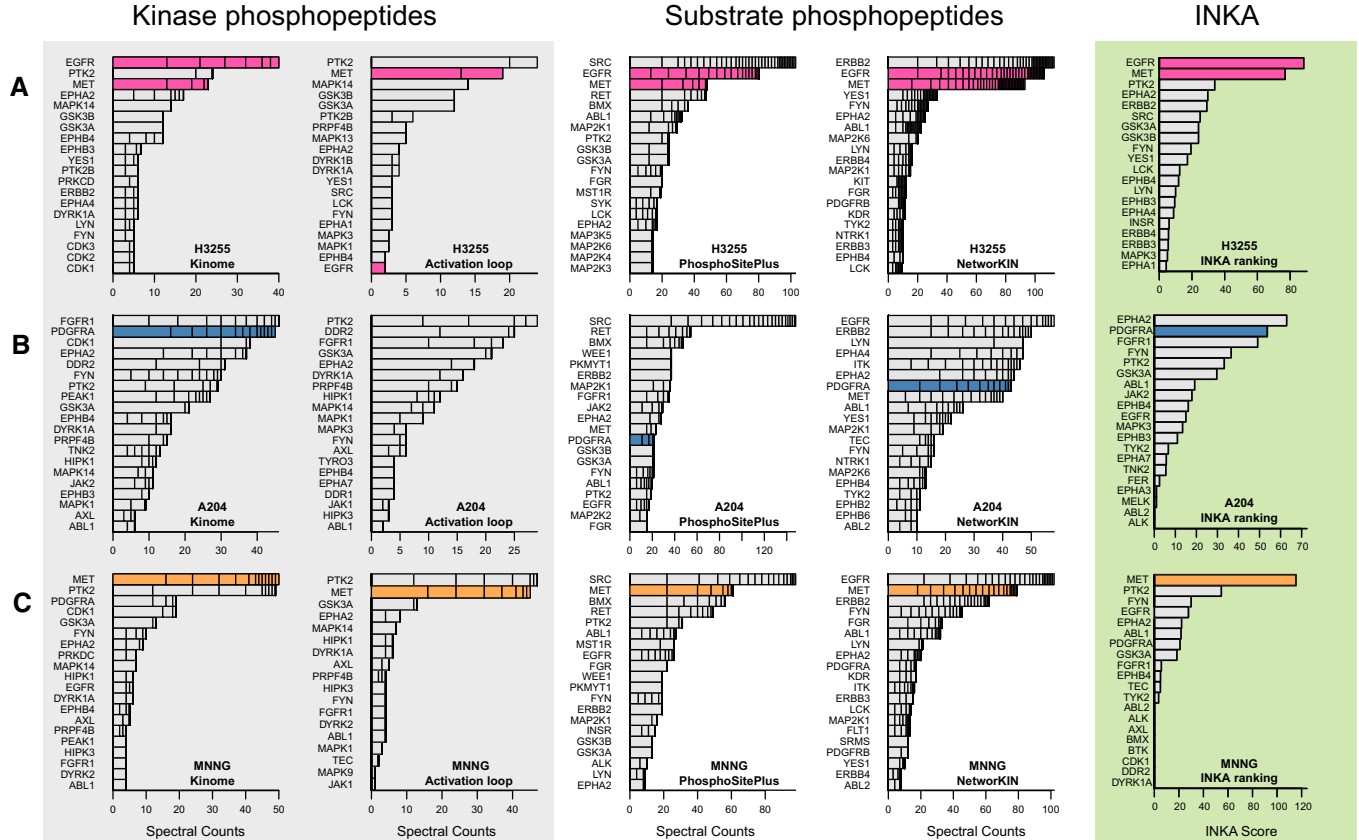

**Figure 4. INKA analysis of three oncogene-driven cancer cell line from the literature.**

A  H3255 NSCLC cells with an *EGFR* mutation. INKA analysis shows that EGFR is a hyper-activated kinase (top 2 in all branches), together with MET.

B  A204 rhabdomyosarcoma cells with documented PDGFRA signaling. PDGFRA exhibits variable ranking (top to intermediate) in individual analysis types for A204, but integrated INKA analysis (right-most bar graph) infers it as a highly active, rank-2 kinase, after EPHA1.

C  MNNG/HOS osteosarcoma cells with documented MET signaling. MET consistently ranks among the top 2 in all analysis arms, culminating in a first rank in the integrative INKA analysis.

Data information: See the legend of Fig 2 for basic explanation.

and F). As expected in the context of DNA damage signaling, INKA scores for ATM and PRKDC/DNA-PK exhibited a time-dependent increase after addition of neocarzinostatin (Fig 5E and F, and Appendix Fig S10). Moreover, the ATM INKA score was significantly reduced after the addition of ATM inhibitor KU55933 (Fig 5E). INKA scoring suggested that the inhibitor influences PRKDC as well (Fig 5F).

To expand application of INKA, we added a script to import phosphoproteomics data that include isobaric (tandem mass tag, TMT) labeling. The publicly available 11-plex TMT dataset PXD009477 of ALK signaling in the neuroblastoma cell line NB1 (Emdal et al, 2018) was re-analyzed using INKA. Upon inhibition of NB1 by ALK-targeting siRNA or the ALK inhibitor lorlatinib, a reduction in the INKA score of ALK by 50% was observed (Appendix Fig S14). Additionally, INKA scores of MAPK pathway members MAP2K1, MAP2K2, MAPK1, and MAPK3 were reduced, indicating ALK-MAPK network crosstalk. Other kinases, including AKT2, RPS6KB1, and RET, showed similar behavior.

In summary, the above differential analyses of phosphoproteomes show that the INKA pipeline can pinpoint target activation

and inhibition after perturbation in both cell lines and clinical samples.

## Comparing INKA to its components and KARP in relation to drug efficacy

To assess the global correlation of INKA with cell line drug efficacy data, we analyzed our cell line use cases (Fig 2) along with two cancer cell panels, with publicly available label-free phosphoproteomics data (Piersma et al, 2015; van der Mijn et al, 2015; Humphrey et al, 2016) and associated drug IC50 data with drug–kinase relationships (Karaman et al, 2008; Davis et al, 2011; Iorio et al, 2016; Klaeger et al, 2017). To this end, we developed an algorithm that takes into account drug IC50 values for different cell lines and provides a global metric for association of drug efficacy and kinase activity ranking. We asked which kinase ranking tool, INKA, its four components, or KARP, correlates kinase activities best with drug efficacy data. Both TiO2 capture data (CRC, PXD001550; Piersma et al, 2015) and pTyr antibody capture data (pancreatic ductal adenocarcinoma, PDAC; PXD003198; Humphrey

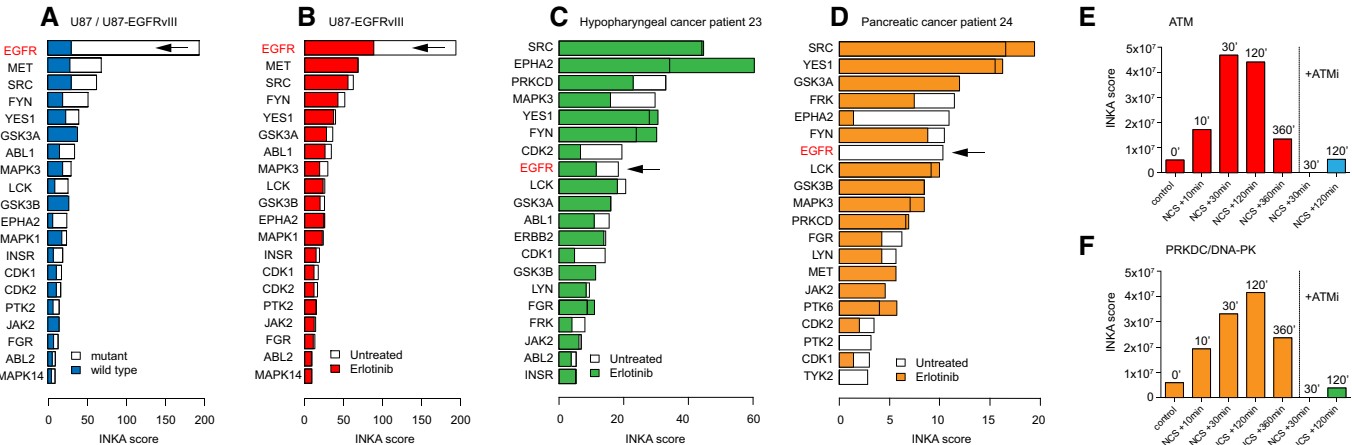

**Figure 5. INKA analysis in differential genetic and pharmacological settings.**

A   Effect of a monogenetic change in a cancer cell line use case. Comparison of U87 glioblastoma cells ("wild-type") with isogenic U87-EGFRvIII cells overexpressing a constitutively active EGFR variant ("mutant") grown under baseline conditions.

B   Effect of drug treatment in a cancer cell line use case. Comparison of U87-EGFRvIII cells at baseline with U87-EGFRvIII cells treated with 10 μM erlotinib for 2 h shows a clearly reduced INKA score for EGFR.

C   Effect of drug treatment in a patient with hypopharyngeal cancer. Tumor biopsies were taken both before and after 2 weeks of erlotinib treatment.

D   Same as panel (C), but for a patient with pancreatic cancer.

E   Time-dependent effect of radiomimetic treatment in a cancer cell line use case. MS intensity-based INKA analysis of TiO$_2$-captured phosphoproteomes from G361 melanoma cells at different time points following treatment with the DNA damage-inducing drug neocarzinostatin (NCS) in the absence or presence of ATM inhibitor KU55933 (ATM). Plotted is the INKA score for ATM, exhibiting a time-dependent increase, which is not observed with ATM blocking. Full INKA score bar graphs are shown in Appendix Fig S10. As these data were generated using an LTQ-FT mass spectrometer that is less sensitive than current orbitrap-based instruments, spectral count data did not work well, and MS intensity data were analyzed instead.

F   Same as panel (E), but plotting of the INKA score for PRKDC/DNA-PK, which exhibits similar behavior.

Data information: Raw data for panels (A and B) are from van der Mijn *et al* (2014). Raw data for panels (E and F) are from Bensimon *et al* (2010) and averaged for replicate treatment conditions.

*et al*, 2016) were included. First, for each method, kinase scores were divided by their maximum value, resulting in a normalized kinase activity score (KAS) for each of the ranked kinases. Second, for each cell line, drug −logIC50 values ($e_j$; Dataset EV6) were median-shifted to zero and proportionally scaled to range from −1 to 1 for the lowest and the highest affinity drugs, respectively. Third, a table of drug–kinase relationships was populated using GDSC (Iorio *et al*, 2016), CCLE (Karaman *et al*, 2008; Davis *et al*, 2011), and Proteomics DB (Klaeger *et al*, 2017; Dataset EV6). Next, for each kinase in the top-N activity rank the normalized kinase activity score is multiplied by $e_j$ of the drugs that are associated with the kinase as specified in the binary interaction table $g_{ij}$. The summation over kinases in the top-N activity rank list yields the combined kinase impact score for a cell line. In Fig 6A, the algorithm for calculation of the kinase impact score is shown. In Fig 6B and D, the kinase impact score for pTyr-based and TiO$_2$-based phosphoproteomics data is shown as a function of the top-N kinase activity list for all cell lines analyzed. The *P*-value reported is the median of *P*-values comparing kinase impact scores in cell lines for INKA and KARP (Mann–Whitney test). In Fig 6C and E, the kinase impact score for the top 10 kinases is shown for INKA, KARP, and INKA components. The kinase impact score is globally higher for INKA than for KARP or for the four INKA components, indicating that INKA is superior in ranking kinase activities in the context of drug efficacy. This is a first step in the application of kinase activity ranking to drug selection.

**INKA analysis reveals kinase drug targets in colorectal cancer patient-derived xenograft tumors**

To explore the drug target potential of kinase with high INKA score, we performed phosphoproteomics and INKA analysis of two patient-derived xenograft (PDX) tumors, CRC0177 and CRC0254, from two patients with metastatic colorectal cancer (Fig 7A and Appendix Fig S11A). INKA analysis showed high inferred kinase activity of a group of receptor tyrosine kinases (IGF1R/INSR, EGFR, ERBB2, various ephrin receptors, and MET), which may engage in active cross talk either directly by heterodimerization or indirectly through downstream effectors (van der Veeken *et al*, 2009; Yamaguchi *et al*, 2014). This group of kinases is important in sustaining and initiating cellular proliferation through activation of the MAPK and PI3K/AKT pathways (van der Veeken *et al*, 2009; Yamaguchi *et al*, 2014; Fig 7B). To inhibit these top-ranking kinase activities in organoid cultures, we selected two different kinase inhibitors, BMS-754807 targeting IGF1R/INSR, MET, and GSK3B, and afatinib targeting EGFR, ERBB2, and ABL1. The experimentally established target space (nanomolar range) of these two inhibitors is depicted in Fig 7C and mapped onto the INKA bar graph. Importantly, INKA analysis of PDX-derived organoids showed overall consistency of target activity in both models CRC0177 and CRC0254 (Fig 7C). Drug treatment of organoids showed a significant reduction of viability ($P < 10^{-5}$ for all comparisons between control and drug-treated conditions), indicating that both models depend on these kinase pathways (Fig 7D). Single treatment with either

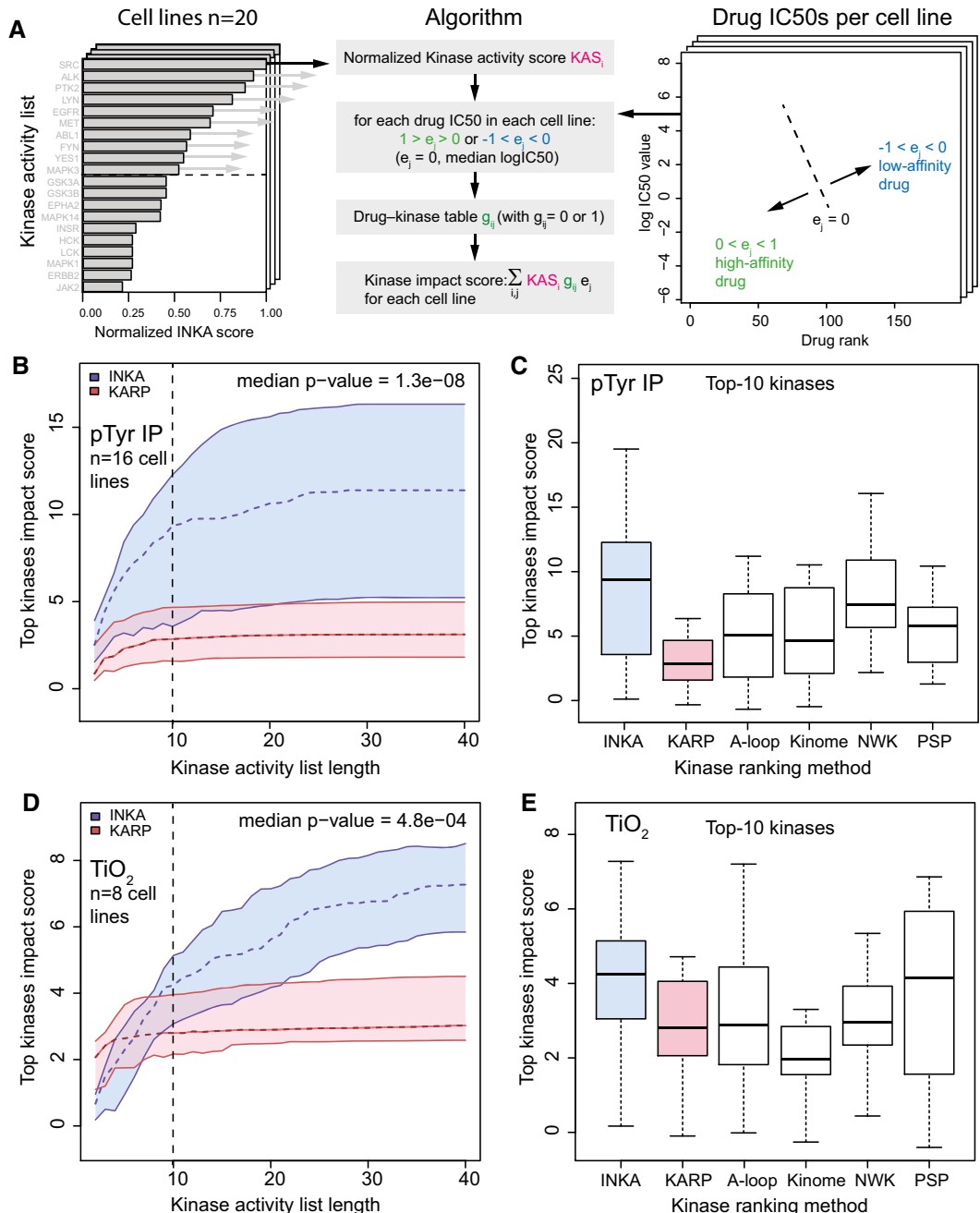

**Figure 6. Kinase activity ranking and drug sensitivity.**

A   Algorithm to calculate kinase impact score from (1) cell line INKA, KARP or INKA components normalized kinase activity ranking $KAS_i$, (2) normalized drug $-logIC50$ values $e_j$, and (3) a drug–kinase relationship table $g_{ij}$.

B   Kinase impact score versus ranked kinase list length for pTyr IP data from PDAC cell lines MIA.PACA2, ASPC1, BXPC3, CAPAN1, CAPAN2, CFPAC1, HPAC, HPAF.II, SU8686, SW1990, HS766T from PXD003198, and cell lines K562, SK-Mel-28, HCC827, H2228, and U87 ($n$ = 16 cell lines, 31 samples), for which drug data were available (dashed line: median value, colored area: interquartile range, INKA: blue and KARP: pink). The *P*-value reported is the median of *P*-values comparing kinase impact scores in cell lines for INKA and KARP (Mann–Whitney test).

C   Kinase impact score for panel (B) for the top-10 kinases shows that the median kinase impact score of INKA > KARP, or INKA components activation loop (A-loop), kinome (kinome), NetworKin (NWK), or PhosphositePlus (PSP).

D   Kinase impact score versus ranked kinase list length for global $TiO_2$ phosphoproteomics data from CRC cell lines COLO205, HCT116, HT29, RKO from PXD001550, and cell lines K562, SK-Mel-28, HCC827, and H2228 ($n$ = 8 cell lines, 20 samples), for which drug data were available (colors and *P*-value calculation: same as in panel B).

E   Kinase impact score for global $TiO_2$ phosphoproteomics data of panel (D) for the top-10 kinases shows that the median kinase impact score of INKA > KARP, or INKA components activation loop (A-loop), kinome (kinome), and NetworKin (NWK). The PhosphositePlus (PSP) component shows a similar kinase impact score as INKA.

Data information: (C, E) Boxplots are Tukey boxplots (box bounded by upper and lower quartile values, with the thick horizontal line inside indicating the median; whiskers extend to the most extreme data point that is still within 1.5 times the interquartile range of the lower and upper quartiles, respectively).

                                                 

BMS-754807 or afatinib resulted in a strong reduction of cellular viability for both CRC0177 ($IC50_{BMS-754807}$ = 25 nM, 95% CI = 18.2–29.5 nM; $IC50_{afatinib}$ = 50 nM, 95% CI = 33.9–64.6 nM) and CRC0254 ($IC50_{BMS-754807}$ = 4 nM, 95% CI = 2.8–5.0 nM; $IC50_{afatinib}$ = 4 nM, 95% CI = 2.8–6.6 nM; Appendix Fig S11B). Interestingly, combination of afatinib and BMS-754807 further increased the detrimental effects on organoid viability ($P < 0.05$ for the drug combination compared to either single-drug treatment).

To explore whether a kinase with a low INKA score does not show a response to the corresponding drug, we selected ABL that ranked low in both PDXs and organoids of CRC0177 and CRC0254. Indeed, organoid treatment with the ABL inhibitor imatinib yielded negligible inhibition (IC50 imatinib = 4 or 6 μM for CRC0177 and CRC0254, respectively; Fig 7D) while the positive control (CML cell line K562) worked (Appendix Fig S11E), underscoring the value of INKA ranking for drug response prediction. CRC0254 was slightly more responsive than CRC0177 for both inhibitors. The lower sensitivity of CRC0177 as compared to CRC0254 for BMS-754807 and afatinib may be explained by the relatively higher AKT activity in this model, as uncovered by INKA analysis of $TiO_2$ global phosphoproteomic data (Appendix Fig S11C and D).

Taken together, our analysis of drug intervention in clinically relevant PDX tumor models indicates that the INKA algorithm has the ability to guide prioritization of oncogenic kinases as drug target candidates.

## Discussion

Present-day cancer treatment is increasingly shifting toward individualized therapy by specific targeting of hyperactive kinases in patient tumors. In this context, with heterogeneity and plasticity of kinase signaling in a specific tumor at a specific time, it is essential to have an overview of hyperactive kinases and prioritize ones that (help) drive malignancy to maximize therapeutic success and minimize expensive failures and unnecessary burden for the patient. Here, we present a novel pipeline, Integrative Inferred Kinase Activity (INKA) scoring, to investigate phosphoproteomic data from a single sample and identify hyperactive kinases as candidates for (co-)targeting with kinase inhibitors. In a first demonstration of its application, we have performed INKA analyses of established cancer cell lines with known oncogenic drivers. We analyzed both data from tyrosine phosphoproteomics in our laboratory and similar data described in the literature. Furthermore, INKA could distinguish relevant differences between closely related mutant and wild-type cells and reveal drug perturbation effects in both tyrosine and global ($TiO_2$) phosphoproteomics data. We also applied INKA scoring to tumor needle biopsies of two patients before and after kinase inhibitor treatment and demonstrated its use for drug target selection in patient-derived xenograft tumors and functional analysis of corresponding organoid cultures.

This study shows that label-free MS-based phosphoproteomics data can be used in a multipronged analysis to infer and rank kinase activity in individual biological samples. Several approaches have been developed for kinase activity inference. In kinase-centric analyses, phosphorylation of the kinase itself (either considering all sites or focusing on the ones located in the activation loop) is used as a proxy for its activation. Activation loop analyses inherently suffer from a focus on a small protein segment. In substrate-centric analyses; instead, substrate phosphorylation is used to deduce kinase activity indirectly through kinase–substrate relationships (based on either experimental knowledge in PhosphoSitePlus or motif-based prediction by NetworKIN). Here, we present INKA, which combines both kinase-centric and substrate-centric evidence in a stringent meta-analysis to yield an integrated metric for inferred kinase activity. The results of INKA highlight kinases that are in line with known cancer biology and show that INKA scoring clearly outperforms substrate-centric analyses alone, and also holds a slight edge over phosphokinase ranking pioneered by Rikova *et al* (2007) and over KARP (Wilkes *et al*, 2017). In particular, kinases expressed from amplified genes or fusion genes that drive tumor growth rank high in INKA scoring, illustrating the power of applying an integrative analysis to in-depth phosphoproteomics data. Furthermore, using public phosphoproteomics data of two cancer cell panels, we showed that INKA is superior in ranking kinase activities in the context of drug efficacy.

Meaningful substrate-centric inference of kinase activity is pivotal to INKA scoring. The PhosphoSitePlus- and NetworKIN-based

---

**Figure 7. INKA-guided selection of kinase drug targets in patient-derived xenograft (PDX) tissue analyzed by pTyr-based phosphoproteomics.**

A   INKA score ranking of the top 20 active phosphokinases in two PDX models (metastatic colorectal cancer, CRC0177 and CRC0254) indicates high activity of receptor tyrosine kinases IGF1R/INSR, EGFR, ERBB2, EPHA1/2, and EPHB1-4, and MET. Bar colors indicate targets of tyrosine kinase inhibitors BMS-754807 (red), afatinib (blue), and imatinib (green). The arrow indicates MET, a receptor tyrosine kinase that is in the INKA top 20 for CRC0254, but not CRC0177.

B   Scheme indicating signal transduction pathways of IGF1R/INSR, EGFR, ERBB2, EPHA2, and MET. These receptors reciprocally activate each other due to physical association and receptor trans-phosphorylation, and induce downstream activation of MAPK and PI3K/AKT signaling with subsequent stimulation of cellular proliferation and survival.

C   Overview of experimentally established targets of BMS-754807 (coral-colored panel), afatinib (blue panel), and imatinib (green panel) with an affinity in the nanomolar range. Targets denoted by circles were discovered using a chemical proteomics approach (Klaeger *et al*, 2017), and those denoted by triangles were identified using cell-free assays (Carboni *et al*, 2009; Mulvihill *et al*, 2009; Davis *et al*, 2011). Kinases in bold type are intended drug targets while kinases that are off-targets are given in plain type. Kinases with a top 20 INKA score for models CRC0177 or CRC0254 are indicated by matching colors as in panel A (coral: BMS-754807, blue: afatinib, green: imatinib).

D   Viability of PDX-derived organoids in response to target inhibition. Organoids were incubated without drugs (black bars), 50 nM INSR/IGF1R inhibitor BMS-754807 (coral-colored bars), 50 nM EGFR inhibitor afatinib (blue bars), both drugs (gray bars), or 50 nM ABL inhibitor imatinib (green). Treatment with individual drugs resulted in a significant reduction of organoid viability, with the combination of both inhibitors leading to almost complete abrogation of cell viability in CRC0254. The increased sensitivity of CRC0254 relative to CRC0177 may be explained by BMS-754807-mediated inhibition of MET, which was inferred to have a high activity in CRC0254 but less so in CRC0177 (see panel A). Imatinib, which targets ABL, a kinase ranking low by INKA for the CRC PDX models and organoids, was used as negative control and indeed showed negligible inhibition. Statistical analysis was performed using ordinary one-way ANOVA. Error bars represent SEM. Asterisks represent the level of significance (**$P < 0.01$; ****$P < 0.0001$). Numbers of replicates analyzed are: for control, 47 (CRC0177) and 17 (CRC0254); for BMS-754807: 16 and 12; for afatinib: 13 and 3; for afatinib+BMS-754807: 6 and 3; for imatinib: 6 and 6.

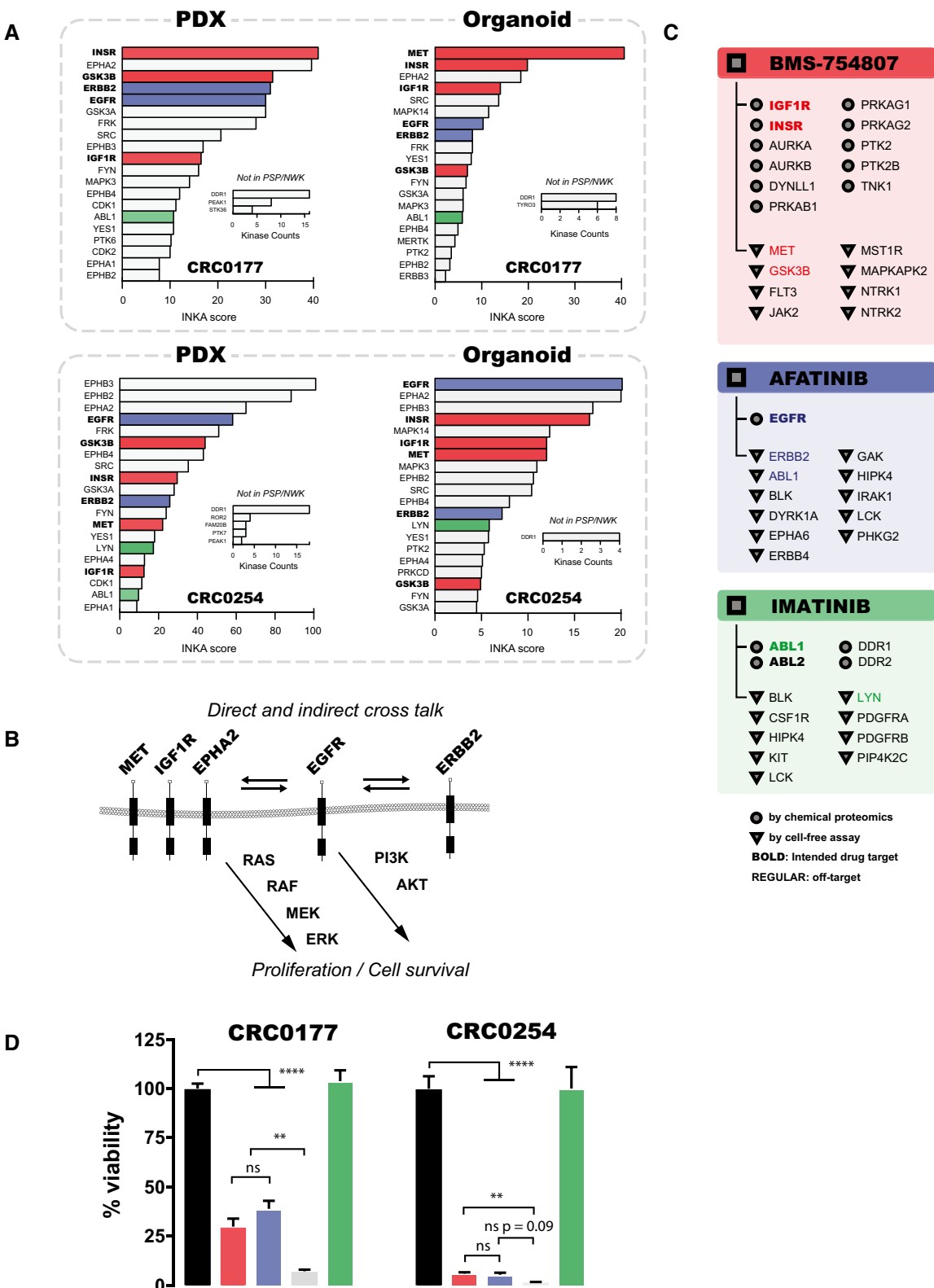

**Figure 7.**

approaches depend on the availability of comprehensive curated data on experimentally observed kinase–substrate relationships or reliable predictions thereof, respectively. To date, only about two-thirds of the kinome has been covered, and more substantial inferences could be made with further population of resources (such as PhosphoSitePlus), especially when the latter also cover cancer-associated aberrations such as fusion gene products BCR-ABL and EML4-ALK (Medves & Demoulin, 2012; Lee *et al*, 2017). To overcome current limitations in information, our INKA pipeline provides a separate kinase-centric ranking of kinases that are not yet covered by PhosphoSitePlus and NetworKIN, but do show up as phosphoproteins in a sample. This reduces the chance of missing important kinases, as illustrated by the case of AXL in the HCC827-ER3 cell line. Moreover, the INKA pipeline generates network visualizations of all kinase–substrate relationships inferred for the top 20 kinases in an experiment. This provides a more instructive overview of phosphoproteomic biology in a sample than mere scoring and ranking alone. Additionally, when analyzing INKA scores of different experiments or laboratories, INKA normalization on the maximum INKA score may standardize scores and allow comparison of datasets.

A next stage is to apply INKA to more advanced cancer models and, especially, clinical samples. To analyze limited amounts of patient tumor tissue in a clinical practice setting, we have recently downscaled tyrosine phosphoproteomics to clinical needle-biopsy levels (Labots *et al*, 2017). Using this workflow, we showed the feasibility of phosphoproteomics combined with INKA analysis of patient samples from a clinical molecular profiling study with kinase inhibitors (Labots *et al*, in preparation), showing a reduced INKA score of EGFR upon erlotinib treatment. Finally, analysis of colorectal PDX and organoids tumors revealed the functional importance of kinases with high INKA rank over a kinase with a low rank.

In summary, INKA scoring can infer and rank kinase activity in a single biological sample, display differences between closely related yet genetically distinct cells and in cells and tissues after drug intervention, and, importantly, identify functionally relevant activated drug targets. Therefore, its application may be much broader than the cancer context. We envision that INKA analysis of phosphoproteomic data on tumor biopsies collected in kinase inhibitor trials can pave the way for future clinical application. The ultimate goal would be tailoring treatment selection for the individual patient.

# Materials and Methods

### Reagents and Tools table

| Reagent/Resource | Reference or Source | Identifier or Catalog Number |
|---|---|---|
| **Experimental models** | | |
| HCC827 cells (*H. sapiens*) | ATCC | Cat# CRL-2868 |
| NCI-H2228 cells (*H. sapiens*) | ATCC | Cat# CRL-5935 |
| SK-Mel-28 cells (*H. sapiens*) | ATCC | Cat# HTB-72 |
| K562 cells (*H. sapiens*) | ATCC | Cat # CCL-243 |
| HCC827-ER3 cells (*H. sapiens*) | Zhang *et al* (2012) | N/A |
| CRC0177 patient-derived xenograft (*H. sapiens* in *M. musculus*) | Bertotti *et al* (2015) | N/A |
| CRC0254 patient-derived xenograft (*H. sapiens* in *M. musculus*) | Bertotti *et al* (2015) | N/A |
| **Antibodies** | | |
| PTMScan® Phospho-Tyrosine Rabbit mAb (P-Tyr-1000) Kit | Cell Signaling Technology | Cat# 8803 |
| **Chemicals, enzymes and other reagents** | | |
| Cultrex PathClear Reduced Growth Factor Basement Membrane Extract, Type 2 | R&D Systems | Cat# 3533-005-02 |
| CellTiter-Glo Luminescent Cell Viability Assay | Promega | Cat# G7570 |
| Sequencing Grade Modified Trypsin | Promega | Cat# V5117 |
| Trypsin Resuspension Buffer | Promega | Cat# V542 |
| PTMScan Phospho-Tyrosine Rabbit mAb (P-Tyr-1000) Kit | Cel Signaling Technology | Cat# 8803 |
| **Software** | | |
| R v3.2.3 - implemented with | http://www.r-project.org | |
| R package data.table v1.10.4 | http://r-datatable.com | |
| R package splitstackshape v1.4.2 | http://github.com/mrdwab/splitstackshape | |
| R package stringr v1.2.0 | http://stringr.tidyverse.org, http://github.com/tidyverse/stringr | |
| R package network v1.13.0 | http://statnet.org | |

**Reagents and Tools table**  (continued)

| Reagent/Resource | Reference or Source | Identifier or Catalog Number |
|---|---|---|
| R package gplots v3.0.1 | http://CRAN.R-project.org/package=gplots | |
| R package tools v3.2.3 | https://www.rdocumentation.org/packages/tools/versions/3.2.3 | |
| **Other** | | |
| Q Exactive Hybrid Quadrupole-Orbitrap Mass Spectrometer | Thermo Fisher Scientific | – |
| Branson Sonifier Model S450 Digital | VWR International | Cat# 432-4503 |
| Branson Cup horn 1″ | VWR International | Cat#142-3743 |
| Protein LoBind Eppendorf microcentrifuge tubes | VWR International | Cat# 525-0133 |
| OASIS HLB Cartridges (6 cc, 500 mg Sorbent, 60 μm Particle Size) | Waters | Cat# 186000115 |
| Empore Styrene Divinyl Benzene (SDB-XC) Disks | Sigma Aldrich | Cat# 66884-U |

## Methods and Protocols

### Cell line culture

For in-house phosphoproteomics, the cell lines in Table 1 were used. H2228 and SK-Mel-28 were cultured in DMEM supplemented with 10% fetal bovine serum and 2 mM L-glutamine. The other cell lines were cultured in RPMI 1640 containing 2 mM L-glutamine and supplemented with 10% fetal bovine serum (K562 as a suspension culture). Cell lines tested negative for mycoplasma.

### Human patient-derived xenograft (PDX) tumor samples

The patient-derived xenograft tumor tissue used in this study was derived from xenograft models that have been described before (Bertotti *et al*, 2015). Tissue was processed as in "Tissue lysate preparation for phosphoproteomics" described below.

### Organoid culture and viability testing

Organoids are grown and tested according to established procedures (van de Wetering *et al*, 2015; Verissimo *et al*, 2016).

- Cut tumor tissue into small pieces, wash with PBS, and dissociate further using shear distress (pipetting).
- Spin down suspension, resuspend pellet in basement membrane extract hydrogel (Cultrex pathClear, RGF BME, type 2)
- Pipette 100 μl as a drop in a well of a 24-well culture plate and allow to solidify at 37°C.

- Culture in DMEM F12 medium supplemented with 1% penicillin/streptomycin, 1% B27, 1% N2, 2 mM L-glutamine, 1 nM N-acetyl-cysteine, and 0.02 μg/ml EGF, refreshing medium every 4 days.
  ○ For splitting, dissociate organoids using shear distress (pipetting), spin down cells, and resuspend in basement membrane extract hydrogel.
  ○ For viability assays:
    – On day 0, prepare single-cell suspensions using shear distress (pipetting) and trypsin-EDTA treatment.
    – Seed cells in multiwell plates coated with basement membrane extract hydrogel and containing the above medium without EGF.
    – On day 1, add drugs.
    – On day 6, assess cell viability with a CellTiter-Glo luminescent cell viability assay.

### Tissue lysate preparation for phosphoproteomics

Tissue is preferentially sliced with a microtome to enhance solubilization; otherwise, a micropestle can be used but may cause losses. Process tissues sequentially, not in a batch.

- Process the desired amount of sliced tissue (enough to yield 5–10 mg protein) by cutting 10- to 20-μm slices of fresh frozen tissue at −20°C in a cryostat.
- Add lysis buffer (9 M urea, 20 mM HEPES pH 8.0, 1 mM sodium orthovanadate, 2.5 mM sodium pyrophosphate, 1 mM β-glycerophosphate) in a 1:40 wet weight-to-lysis buffer ratio.

**Table 1.**  Oncogene-driven cancer cell lines analyzed by phosphoproteomics for this study. ATCC:  American Type Culture Collection.

| Cell line | Source | Cancer type | Aberration/Driver | Reference |
|---|---|---|---|---|
| H2228 | ATCC | Non-small-cell lung cancer | *EML-ALK* fusion | Soda *et al* (2007), Rikova *et al* (2007) |
| SK-Mel-28 | ATCC | Melanoma | *BRAF* V600E mutation | Carey *et al* (1976), Davies *et al* (2002) |
| K562 | ATCC | Chronic myelogenous leukemia | *BCR-ABL1* fusion | Klein *et al* (1976), Heisterkamp *et al* (1985) |
| HCC827 | ATCC | Non-small-cell lung cancer | *EGFR* E746-A750 deletion | Furugaki *et al* (2014) |
| HCC827-ER3 | Dr. B. Halmos, Columbia University Medical Center, New York, USA | Non-small-cell lung cancer | *EGFR* E746-A750 deletion; acquired resistance to erlotinib | Zhang *et al* (2012) |

- Vortex for 30–60 s at room temperature and at maximum speed until all tissue has been solubilized. If the lysate is very viscous, add additional lysis buffer.
- Immediately sonicate in three cycles, e.g., 15 s on/1 min off at maximum amplitude when using a Branson high-intensity cuphorn sonicator.
- Centrifuge in a microcentrifuge for 15 min at maximum speed and at room temperature for samples in 1.5- or 2-ml Eppendorf tubes, or for 30 min at $6,000 \times g$ if the volume is larger.
- Transfer supernatant (cleared lysate) to a fresh Eppendorf tube.
- Take small aliquots for SDS–PAGE quality control and protein concentration determination.
- Snap-freeze lysates in liquid nitrogen and store at −80°C until use.

For tumor biopsy phosphoproteomics in this study, needle biopsies from a patient with advanced head and neck squamous cell carcinoma, obtained in an Institutional Review Board-approved molecular profiling study before and after 2 weeks of treatment with erlotinib (NCT clinical trials identifier 01636908; www.clinicaltrials.gov), were processed as described elsewhere (Labots *et al*, 2017). For phosphoproteomics, a 2.5 mg protein equivalent was used.

### Cell lysate preparation for phosphoproteomics
- Grow cells to 70–80% confluency in 15-cm dishes (adherent cell lines) or to $1 \times 10^6$ cells/ml (suspension cultures). Depending on the cell line, $1$–$3 \times 10^8$ logarithmically growing cells are needed to obtain a protein yield of ∼10–20 mg.
- Lyse cultures batchwise, e.g., for a large set of dishes with the *same* cells:
  ○ Discard culture medium from a set of three dishes, carefully add 5 ml PBS, and swirl dishes briefly.
  ○ Discard PBS and leave dishes in tilted position for 1 min, and then remove remaining PBS with a Pasteur pipette.
  ○ Add 2 ml lysis buffer (9 M urea, 20 mM HEPES pH 8.0, 1 mM sodium orthovanadate, 2.5 mM sodium pyrophosphate, 1 mM β-glycerophosphate) to each of the three dishes and swirl to distribute the buffer over the entire bottom of the dish (cell layer).
  ○ Scrape cells into the lysis buffer with a cell scraper, leave dishes in tilted position to allow lysate to drain to the bottom, wait for a short time, then collect the lysate from the dishes, and return it to a tube labeled with the sample name.
  ○ Repeat the above steps with a second set of three dishes. For lysis, use the *same* "lysis buffer" (already containing lysate from the first three dishes) from the labeled tube prepared in the previous step.
  ○ Continue with three-dish batches until all dishes with the same cells have been processed. The labeled tube now contains the pooled lysate of all culture dishes with the same cells.
- Put a 15-ml tube with lysate on ice *just before sonication* (urea will precipitate out after prolonged cooling). Immediately sonicate in three cycles, e.g., 15 s on/1 min off at maximum amplitude when using a Branson high-intensity cuphorn sonicator.
- Centrifuge sonicated lysate for 15 min at $5,400 \times g$ and 10°C.
- Transfer the supernatant to a new tube and store at −80°C until use.

### For suspension cell cultures
- Divide a volume of cell suspension expected to yield 10–12 mg protein over 50-ml tubes.

- Centrifuge at $550 \times g$ for 5 min at room temperature.
- Carefully remove the supernatant and *very gently* loosen pellet by tapping.
- To wash cells, add 25 ml cold PBS to the first tube, quickly but gently resuspend the cell pellet and transfer the suspension to the next tube, collecting all cells in the same 25 ml of PBS.
- Add 25 ml cold PBS to the 25 ml cell suspension.
- Centrifuge at $550 \times g$ for 5 min at 4°C and remove supernatant.
- Lyse cells at room temperature by adding 6 ml lysis buffer to the pellet, pipetting up and down a few times, and vortexing 30 s at maximum speed.
- Continue with the sonication step described above.

### Digestion and desalting of cell lysates for phosphoproteomics
- Pipette a 10 mg protein equivalent of lysate into a 50-ml tube and dilute with lysis buffer to a protein concentration of 2 mg/ml (i.e., to a final volume of 5 ml).
- Reduce proteins by adding 500 μl 45 mM DTT, mixing well, and incubating in a water bath for 30 min at 55°C.
- Cool to room temperature in ice/water. The tube should not be warm or cold.
- Alkylate proteins at room temperature by adding 500 μl freshly prepared 110 mM iodoacetamide, mixing well, and incubating for 15 min in the dark.
- Dilute the reduced and alkylated lysate fourfold to reduce the urea concentration to 2 M while maintaining a 20 mM HEPES pH 8.0 concentration in the diluted lysate.
- Add 1/200 volume of a 1 mg/ml Sequencing Grade Modified Trypsin solution in Trypsin Resuspension Buffer (final concentration 5 μg/ml) and incubate O/N at room temperature.
- Acidify the digest by adding 1/20 volume 20% TFA (final concentration 1%).
- Invert sample, vortex for 5 s, and incubate on ice for 15 min.
- Check pH using a pH strip. The pH should be < 3, otherwise use more 20% TFA.
- Centrifuge for 5 min at $4,000 \times g$ to pellet precipitates and transfer supernatant to a new tube.
- Connect a 500-mg Oasis HLB (hydrophilic–lipophilic balance) cartridge to a vacuum manifold.
- Pre-wet the HLB sorbent with 6 ml acetonitrile and apply vacuum to achieve a flow rate of 1 drop per second (1–1.5 ml/min).
- Wash three times with 6 ml 0.1% TFA.
- Load the acidified and cleared digest in several portions.
  ○ Collect all remaining liquid from the storage tube by centrifuging for ∼20 s at $1,500 \times g$ and add this to the OASIS HLB cartridge.
- Wash three times with 6 ml 0.1% TFA
- Elute with 6 ml elution buffer (50% acetonitrile, 0.1% TFA) into a glass vial with screwcap.
- Snap-freeze eluate in liquid nitrogen and lyophilize for at least 48 h.

### Immunoprecipitation of phosphotyrosine-containing peptides (pTyr IP)
This protocol describes immunoprecipitation with P-Tyr-1000 phosphotyrosine-specific antibody-conjugated beads (20 μl beads per 5–10 mg sample).

## Aliquoting antibody-coupled beads

- Take the required amount of P-Tyr-1000 antibody beads (40 µl 50%-slurry per IP) from the supplied vial.
  - ○ When requiring more than 80 µl slurry (1 supplier vial) to cover all IPs, pool the required volume of slurry into a 1.5-ml LoBind Eppendorf tube as a "stock" vial, and put it on ice.
  - ○ Collect as many remaining beads from the supplier vials as possible by rinsing all empty vials with one volume of 1 ml cold IP buffer (50 mM MOPS pH 7.2, 10 mM sodium phosphate, 50 mM NaCl) and add to the "stock" vial.
- Centrifuge at 2,000 × $g$ for 30 s at 4°C to pellet the beads.
- Carefully remove most supernatant while staying away from the bead pellet, allowing some supernatant to remain on the beads.
- Wash the beads by adding 1 ml cold IP buffer and inverting five times.
- Centrifuge at 2,000 × $g$ for 30 s at 4°C to pellet the beads.
- Carefully remove most supernatant while allowing some supernatant to remain on the beads.
- Repeat the 1 ml IP buffer wash three times. Add 1 ml IP buffer to the final bead pellet.
- Aliquot the beads into as many 1.5-ml LoBind Eppendorf tubes as there are IPs: "IP" tubes.
  - ○ For faithful aliquoting of equal amounts of beads, it is essential to apply a stepwise strategy:
    - – invert the "stock" vial with beads 10 times and distribute a fraction (e.g., 1/4) of the "stock" to the "IP" tubes—restore the volume of the "stock" vial with IP buffer;
    - – perform a second distribution round by again distributing (only) a fraction of the "stock" and restoring the volume of the "stock" vial with IP buffer, etc.;
    - – in a final round (e.g., round 4) completely distribute the contents of the "stock" vial, emptying it. In the end, each "IP" tube should have received 20 µl beads.
- Centrifuge the "IP" tubes with aliquoted beads at 2,000 × $g$ for 30 s at 4°C.
- Carefully remove supernatant, allowing a small volume of IP buffer to remain on the beads.
- Visually check equal distribution of the beads to the "IP" tubes and store on ice.

## Redissolving lyophilized lysate digest

- Add 700 µl IP buffer to each glass vial with lyophilized peptides and dissolve peptides for 5 min, resuspend by gently pipetting up and down, and transfer to a 1.5-ml LoBind Eppendorf tube.
  - ○ Optional: centrifuge the glass vial at 1,000 × $g$ and transfer leftovers to the Eppendorf tube.
  - ○ Optional: spike in a phosphopeptide standard.
- Using a _minimal_ volume of peptide solution (≤ 5 µl), check for a pH ≥ 6 with a pH strip. If the pH is below 6 (remaining TFA), neutralize with 5–10 µl 1 M Tris.
- Centrifuge for 5 min at 16,000 × $g$ and room temperature to pellet all particulate matter. Cool the tube on ice.
  - ○ A 5 µl sample can be taken for checking purposes.

## Immunoprecipitating peptides

- Transfer the supernatant from the previous spin (peptide solution) to an "IP" tube with 20 µl aliquoted antibody-conjugated beads, pipetting directly on top of the beads (without touching).
  - ○ Spin down any remaining liquid sticking to the walls of the peptide solution tube and add that to the peptide/bead mixture.
- Incubate the peptide/bead mixtures for 2 h on a head-over-tail rotator at 4°C in a cold room.
- Centrifuge the mixtures for 30 s at 2,000 × $g$ and 4°C.
- Transfer the supernatant (unbound fraction) to a 1.5-ml LoBind Eppendorf vial for checking and orthogonal purposes. Store at −80°C.
- Remove the last few microliters of supernatant from "IP" tubes with a flat tip, making sure not to touch the beads.
- Add 1 ml cold IP buffer per 10 mg protein input to the bead pellet and mix by inverting five times.
- Centrifuge 30 s at 2,000 × $g$ and 4°C, and remove supernatant, allowing some remaining liquid on the beads.
- Repeat the above washing procedure with IP buffer once.
- Perform three rounds of washing as above but with Milli-Q water instead of IP buffer.
- After the last supernatant removal, centrifuge 5 s at 2,000 × $g$ (to collect all liquid from the walls), and remove all supernatant carefully with a flat tip.
- Elute bound peptides by adding 40 µl 0.15% TFA to the bead pellet and mix by gently flicking the bottom of the tube. Do not vortex.
- Leave for 10 min at room temperature and mix gently every 2–3 min.
- Centrifuge 30 s at 2,000 × $g$, transfer supernatant (eluate 1) to a 1.5-ml LoBind Eppendorf tube.
- Perform another round of elution by adding 30 µl 0.15% TFA to the bead pellet. Following incubation and centrifugation, transfer the supernatant (eluate 2) to the tube with eluate 1.
- Rinse the beads by adding an additional 30 µl 0.15% TFA at room temperature, and immediately retrieve 20–25 µl by careful pipetting, making sure not to take along any beads. Add the wash to the tube with eluate 1 and eluate 2 and mix.

## StageTip desalting and LC preparation of immunoprecipitated peptides

Reverse-phase StageTips are made by punching 1-mm round plugs from Empore Solid Phase Extraction Disk material harboring SDB-XC (styrene/divinylbenzene cross-linked copolymer) as a non-silica-based resin, and lodging the plug at the narrow end of a 20-µl pipette tip.

- Activate the SDB-XC resin by adding 20 µl elution solution (50% acetonitrile, 0.1% TFA) and centrifuge for 1 min at 1,000 × $g$.
- Equilibrate the resin by adding 20 µl 0.1% TFA and centrifuge for 1 min at 1,000 × $g$.
- Load peptide sample in the StageTip and centrifuge for 3 min at 1,000 × $g$.
- Wash by adding 20 µl 0.1% TFA and centrifuge for 1 min at 1,000 × $g$.
- Fit the StageTip on a glass-lined autosampler vial (use a custom-made adapter).
- Elute bound peptides by adding 20 µl elution solution to the StageTip and centrifuge the assembly for 2–5 min in a low-speed centrifuge with a rotor that can hold it.
- Prior to LC-MS/MS, dry down the eluate (now in the autosampler vial) in a vacuum centrifuge to remove the acetonitrile originating from the elution solution.

- Redissolve peptides in 20 μl loading solvent for LC (4% acetonitrile, 0.5% TFA), pipetting 10 times up and down.

### LC-MS/MS

Peptides are separated on an Ultimate 3000 nanoLC-MS/MS system (Dionex LC-Packings, Amsterdam, The Netherlands) equipped with a 20-cm, 75-μm inner diameter fused silica column, custom packed with 1.9-μm ReproSil-Pur C18-AQ silica beads (120-Å pore size; Dr. Maisch, Ammerbuch-Entringen, Germany). After injection, peptides are trapped at 6 μl/min on a 10-mm, 100-μm inner diameter trap column packed with 5-μm ReproSil-Pur C18-AQ silica beads (120-Å pore size) in buffer A (buffer A: 0.5% acetic acid, buffer B: 80% acetonitrile, 0.5% acetic acid) and separated at 300 nl/min with a 10–40% buffer B gradient in 90 min (120 min inject-to-inject). Eluting peptides are ionized at a potential of +2 kV and introduced into a Q Exactive mass spectrometer (Thermo Fisher, Bremen, Germany). Intact masses are measured in the orbitrap with a resolution of 70,000 (at m/z 200) using an automatic gain control (AGC) target value of $3 \times 10^6$ charges. Peptides with the top 10 highest signals (charge states 2+ and higher) are submitted to MS/MS in the higher-energy collision cell (4-Da isolation width, 25% normalized collision energy). MS/MS spectra are acquired in the orbitrap with a resolution of 17,500 (at m/z 200) using an AGC target value of $2 \times 10^5$ charges and an underfill ratio of 0.1%. Dynamic exclusion is applied with a repeat count of 1 and an exclusion time of 30 s.

### Peptide identification and quantification

MS/MS spectra are searched against theoretical spectra based on a UniProt complete human proteome FASTA file (release January 2014, no fragments; 42104 entries) using MaxQuant 1.4.1.2 software (Cox & Mann, 2008). Enzyme specificity is set to trypsin, and up to two missed cleavages are allowed. Cysteine carboxamidomethylation (+57.021464 Da) is treated as fixed modification and serine, threonine and tyrosine phosphorylation (+79.966330 Da), methionine oxidation (+15.994915 Da) and N-terminal acetylation (+42.010565 Da) as variable modifications. Peptide precursor ions are searched with a maximum mass deviation of 4.5 ppm and fragment ions with a maximum mass deviation of 20 ppm. Peptide and protein identifications are filtered at a false discovery rate of 1% using a decoy database strategy. The minimal peptide length is set at 7 amino acids, the minimum Andromeda score for modified peptides at 40, and the corresponding minimum delta score at 17. Proteins that cannot be differentiated based on MS/MS spectra alone are clustered into protein groups (default MaxQuant settings). Peptide identifications are propagated across samples using the "match between runs" option. For the data in Fig 2, the average number of datapoints over the eluting peak is 31. Phosphopeptide MS/MS spectral counts (Liu *et al*, 2004) are calculated from the MaxQuant evidence file using R.

### INKA analysis

The INKA analysis pipeline is implemented in R, utilizing data extracted from web resources (see Table 2). UniProt data (download date June 8, 2016) are used for annotation of UniProt accessions; PhosphoSitePlus data (download date July 3, 2016) are used as experimentally observed phosphosites and kinase–substrate relationships (Phosphorylation_site_dataset and Kinase_Substrate_Dataset, respectively); KinBase data (download date July 20, 2016)

**Table 2. Websites and references for resources used in this study.**

| Resource | Weblink | Reference |
|---|---|---|
| UniProt | www.uniprot.org | UniProt Consortium (2015) |
| PhosphoSitePlus | www.phosphosite.org | Hornbeck *et al* (2015) |
| KinBase | kinase.com/web/current/kinbase | Manning *et al* (2002) |
| HGNC | www.genenames.org | Yates *et al* (2017) |
| Phomics | phomics.jensenlab.org/activation_loop_peptides | Munk *et al* (2016) |

are used as currently recognized protein kinases; HGNC data represent official gene symbols. The "Phomics" tool is used for the annotation of kinase activation loop peptides, and a locally running version of NetworKIN (Horn *et al*, 2014) is used to predict kinases responsible for observed phosphosites. Prior proteome-wide Phomics and NetworKIN analyses are based on a UniProt human reference proteome FASTA file derived from release 2014_01 filtered for "no fragments", and containing 21849 TrEMBL entries and 39703 Swiss-Prot entries. Proteome-wide Phomics analysis entails an upload to the webtool of tryptic peptide sequences generated *in silico* from the FASTA file (March 27, 2017). Proteome-wide NetworKIN analysis entails a batchwise analysis of all proteins in the FASTA file using code from NetworKIN3.0_release.zip downloaded (August 23, 2016) from the NetworKIN website (requiring fixing some code in file pssm_code.h that causes crashing of large FASTA analyses; the fixed and commented code is available at http://inkascore.org/binary_human_public_20140130_ah_fixed.tar.gz). The filtered output only contains kinase predictions ("tree" = KIN) with a networkin_score of ≥ 2.0, which in addition exceeds 90% of the maximum score for a given substrate (substrates with string_identifier ENSP00000376688 are attributed to LYN instead of LCA5).

### Data filtering and annotation

- Phosphopeptide data:
  - Identified and quantified peptides are extracted from modificationSpecificPeptides table.
  - Rows with peptide data that are linked to multiple UniProt gene symbols are deconvoluted into separate rows with a single gene symbol (giving a table called "table 1" here).
- Phosphosite data: the MaxQuant "Phospho (STY)Sites" table is processed:
  - Rows are filtered for so-called class I sites (localization probability > 0.75).
  - Rows linking a phosphosite to multiple UniProt accessions, or to multiple phosphopeptides, are deconvoluted into separate rows.
  - Data from the above web resources are used to annotate all rows in order to prioritize rows that link the same phosphosite to the same gene.
  - Only rows with the best annotated accession for a given phosphosite-gene combination are retained (giving a table called "table 2" here).
- Phosphopeptide data from table 1 and phosphosite data from table 2 are merged in a single, non-redundant class I phosphosite-phosphopeptide table (giving a table called "table 3" here).

### Plot data generation

- A "kinome" analysis data table is generated from table 3 by:
  - Removing redundant rows with different UniProt gene symbols linking the same phosphopeptide to the same HGNC-mapped gene symbol.
  - Filtering for rows with phosphopeptides that are derived from protein kinases.
- An "activation loop" analysis data table is generated by:
  - Identifying kinase-derived phosphopeptides that harbor activation loop sites, utilizing a Phomics data table with the results of a prior proteome-wide Phomics analysis of tryptic peptides (maximum of 2 missed cleavages) predicted *in silico* for the proteins in the above FASTA file.
  - Filtering the "kinome" analysis data table for those phosphopeptides.
- A "PhosphoSitePlus" (PSP) analysis data table is generated by:
  - Merging table 3 with data from the PhosphoSitePlus Kinase_Substrate_Dataset that details experimentally observed human kinase–substrate relationships.
- A "NetworKIN" (NWK) analysis data table is generated by:
  - Using our adapted implementation of the NetworKIN algorithm, in combination with the same FASTA file that was used for peptide and protein identification by MaxQuant, to predict proteome-wide kinase–substrate relationships for phosphosite data from table 3.
  - Filtering prediction results for a NetworKIN score that is not lower than 2 and, in addition, exceeds 90% of the score of the top prediction for the same phosphosite.
  - Merging table 3 with the filtered prediction results.
- For kinase-centric "kinome" and "activation loop" analyses, lists with sample-specific plot data tables are generated by:
  - Creating, for each sample, a plot data table from the above "kinome" or "activation loop" analysis data table with phosphopeptide data.
    - The columns with kinase gene symbols, spectral counts for the pertinent sample, and number of phosphomodifications are extracted from the parent analysis data table.
    - Spectral counts are multiplied by the number of phosphomodifications of the pertinent peptide so as to account linearly for all phosphorylation activity impinging on the substrate protein. This is the "phosphosignal" that is used for a phosphopeptide.
  - Adding the sample-specific plot data table to the list.
- For substrate-centric "PSP" and "NWK" analyses, lists with plot data tables for individual samples are generated by:
  - Performing similar steps as above for kinase-centric analyses, *but*:
    - Extracting data from the above "PSP" or "NWK" analysis data table with coupled phosphosite-phosphopeptide data.
    - Additionally dividing multiplied spectral counts by the number of MaxQuant-inferred phosphosites for a given phosphopeptide and a given kinase–substrate combination. As the latter number may exceed the number of actual phosphomodifications of the phosphopeptide, this makes sure the resulting "phosphosignal" is not exaggerated.

### Kinase bar graph plotting

- Per sample, and per analysis component ("kinome", "activation loop", "PSP", "NWK"), plot data tables are used to produce a top 20 kinase bar graph:

- For substrate-centric analyses, the "phosphosignals" for all phosphopeptides harboring a specific phosphosite are aggregated in the plot data table to obtain a phosphosite "phosphosignal".
- All phosphopeptide "phosphosignals" (kinase-centric analyses), or all phosphosite "phosphosignals" (substrate-centric analyses), attributed to a specific kinase are aggregated.
- The table is filtered for the top 20 kinases with the highest aggregated "phosphosignals".
- A stacked bar graph is plotted where bar segments represent the "phosphosignal" of individual phosphopeptides (kinase-centric analyses) or phosphosites (substrate-centric analyses) contributing to the aggregated value (bar size) for a kinase.

### INKA analysis and plotting

- Per sample, data are extracted from the four separate analysis components ("kinome", "activation loop", "PSP", and "NWK") and integrated:
  - Kinase-specific, aggregated "phosphosignals" from the "kinome" and "activation loop" analyses are summed to get a kinase-centric measure (equation 1).
  - Kinase-specific, aggregated "phosphosignals" from the "PSP" and "NWK" analyses are summed to get a substrate-centric measure (equation 2).
  - An INferred Kinase Activity (INKA) score is calculated as the geometric mean of the two sums from equations 1 and 2 (equation 3).
  - A "skew" parameter is calculated to indicate the relative contribution of kinase-centric versus substrate-centric evidence to the INKA score (equation 4).
    - The parameter is calculated by taking a goniometric correlate of the ratio of substrate-centric and kinase-centric evidence, and normalizing the corresponding angle on $[0,\frac{\pi}{2}]$ to give a value in $[0,1]$. It equals 0 when all evidence is kinase-centric, 1 when all evidence is substrate-centric, and 0.5 when there is equal contribution from both sides

$$C_{kin} = C_{Kinome} + C_{ActivationLoop}, \tag{1}$$

$$C_{sub} = C_{PSP} + C_{NWK}, \tag{2}$$

$$INKA\ score = \sqrt{C_{kin} \cdot C_{sub}}, \tag{3}$$

$$Skew = \frac{2}{\pi} arctan\left(\frac{C_{sub}}{C_{kin}}\right). \tag{4}$$

- A scatter plot is created for kinases observed and/or inferred in the sample, with the INKA score on the vertical axis and the skew parameter value on the horizontal axis, respectively.
  - The plot is limited to kinases with an INKA score of at least 10% of the top INKA score for the sample.
- Flanking the scatter plot, a bar graph is plotted for (maximally) the top 20 "out-of-scope" kinases with a total of at least two spectral counts.
  - For out-of-scope kinases, no experimentally observed substrates are documented (PSP) and no reliable phosphorylation sequence motif for substrate prediction is known (NWK). These kinases will always get a zero $C_{sub}$ score (equation 2), and consequently a zero INKA score (equation 3). To still include

them in the global analysis, a separate ranking is based solely on the kinase-centric side of the INKA score, i.e., $C_{kin}$ (equation 1).

- As a more simplified visualization, INKA scores for the top 20 kinases are also plotted in a bar graph for each sample.

### Kinase–substrate relationship network plotting

- Per sample, a network of inferred kinase–substrate relations annotated with INKA scores is visualized using the R package "network" (Butts, 2008).
  - The "PSP" analysis table and the "NWK" analysis table generated above are merged and relevant data are extracted.
  - Directed edge lists are created, with kinases and substrates as tail and head nodes, respectively.
  - Other annotation data are stored as node and edge attributes in the network-class R object.
  - The final network object is printed.
    - Node coordinates are calculated using a Fruchterman–Reingold layout algorithm for force-directed graph drawing with empirically chosen parameter settings (for N nodes: niter = 100N, area = $N^{1.8}$, repulse.rad = $N^{1.5}$, and ncell = $N^3$).
    - Nodes are depicted as a hexagon (observed kinases, identified through one or more phosphopeptides), a pentagon (inferred kinases lacking direct observation, but linked to phosphorylation of one or more observed phosphopeptides), or as a circle (non-kinase substrates).
    - Kinase node colors correlate with INKA scores in a white to red gradient, and kinases with at least one phosphorylated activation loop phosphosite are indicated by a thicker node border.
    - Edge widths correlate with the associated substrate site "phosphosignal", and edge colors indicate the analysis on which the kinase–substrate relationship was based (coral: PSP, cornflowerblue: NWK, forestgreen: both).

### Statistical significance assessment

- Random INKA scores are generated for each kinase in each sample using a twofold randomization procedure:
  - Experimental data for a sample are randomized:
    - All non-zero spectral count values are permuted.
  - Knowledgebase data are randomized:
    - All kinases in kinase–substrate relations in PhosphoSitePlus and NetworKIN are replaced by a random kinase from the pool of kinases present in kinase–substrate relations
- This procedure is iterated 100,000 times to give sample- and kinase-specific null distributions.
- Using the null distributions, *P*-values are calculated for each of the original INKA scores associated with kinases for a given sample.

### Kinase impact score calculation

Performance of INKA and KARP was compared by calculation of the top kinase impact score. This score quantitatively integrates the ranking of kinases in a kinase activity score list, with measured efficacy of kinase inhibitors in cell lines. To this end, we compiled suitable and publicly available cell line phosphoproteomics data. First, a pTyr IP cell line dataset derived from PDAC cell lines (Humphrey *et al*, 2016), available from proteomeXchange (PXD003198), was composed and supplemented with pTyr data from HCC827, SK-Mel28, K-562, and H2228 cell lines used in this manuscript, with additional pTyr data of a

U87 cell line obtained from proteomeXchange (PXD001565; van der Mijn *et al*, 2015). Second, a $TiO_2$ enrichment-based global phosphoproteomics dataset was assembled from data published by Piersma *et al* (2015) (proteomeXchange PXD001550) and supplemented with $TiO_2$ data for HCC827, SK-Mel28, K-562, and H2228 cell lines from the current study. Furthermore, we implemented the KARP algorithm from the Cutillas group (Wilkes *et al*, 2017) in R, to calculate KARP scores for all cell lines, while the remaining score lists were generated using our INKA software.

IC50 values for the cell lines were obtained from the GDSC website (Iorio *et al*, 2016) (version 17.3). Kinase drug targets from the GDSC dataset were supplemented with compiled drug targets from the literature (Karaman *et al*, 2008; Davis *et al*, 2011; Klaeger *et al*, 2017). Retaining only drugs that target at least one kinase resulted in a table linking cell lines, drug IC50 values, and targeted kinases (Dataset EV6). The pTyr dataset contained in total 31 phosphoproteomics datasets, covering 16 different cell lines that could be linked to drug IC50 data, whereas the $TiO_2$ cell line panel contained 20 different samples from 8 cell lines, linked to IC50 data. On average, more than 110 IC50 values for kinase-targeting drugs could be attributed per cell line sample. Cell line-specific details can be found Dataset EV6.

The top kinase impact score for each cell line and kinase activity score list was calculated from an INKA (or KARP) analysis of MaxQuant search results. To eliminate scale effects stemming from method-specific score magnitude differences, all score lists were normalized by division of all scores by the highest score in the list. Next, all logIC50 values for each cell line under consideration were transformed by subtraction of the median, multiplication by minus one, and division by the maximum absolute value. This resulted in all drugs having normalized −logIC50 values between −1 and 1, with high-affinity drugs having positive and low-affinity drugs having negative values. Subsequently, a summation over all kinases in the list and drugs targeting these kinases (via the drug–kinase table $g_{ij}$, Dataset EV6) was performed of the product of the drug-normalized IC50 times the normalized kinase activity score. This score rewards high-ranking kinases that are targeted by predominantly high-affinity drugs while penalizing kinases associated with low-affinity drugs. This procedure was performed on both pTyr and $TiO_2$ datasets separately for all lengths of kinase activity lists ranging from 2 to 40 kinases. Next, for each length of INKA and KARP kinase activity lists, we performed a Mann–Whitney test paired over cell lines to assess the statistical significance of the difference between the two kinase activity ranking methods, summarizing the difference by the median of these *P*-values.

### INKA: spectral counts versus Intensity

Although INKA analysis can be performed with intensity-based quantification, we favor spectral count-based quantification as it is less sensitive to peptides with outlier intensities and is more robust for the analysis of aggregated data for multiple peptides, some of which may exhibit dominantly high intensities. For Q Exactive data, spectral counting outperformed intensity-based quantification for INKA-based kinase ranking of known drivers (Appendix Fig S12), yet for the low-level LTQ-FTMS data, the intensity data worked better (Fig 5E and F).

### iBAQ correction for INKA

We explored implementation of an iBAQ procedure (Schwanhäusser *et al*, 2011) to correct for the number of phosphopeptides per kinase

and the number of substrates per kinase. For the substrate side of INKA, we divided the PSP and NWK spectral counts of each kinase, by the number of kinase–substrate relations present in the respective kinase–substrate networks for the kinase under consideration. On the kinase side of INKA, kinome counts were divided by the number of kinase peptides that contained an amino acid that could be phosphorylated and were at least 7 amino acids long. For pTyr IP experiments, at least one tyrosine should be present in the peptide, whereas for $TiO_2$ experiments, also peptides containing at least a serine or threonine were taken into account. The activation loop peptide contribution was left unchanged. For the eight samples considered (four cell lines of Fig 2 and the four U87 conditions of Fig 5A and B), the highest rank for the driver kinases was either equivalent or better for uncorrected INKA score calculations. Therefore, we decided not to incorporate the iBAQ approach into INKA; see also Appendix Fig S13.

*INKA analysis of 11-plex isobaric TMT data*

We adapted INKA to phosphoproteomics data with 11-plex TMT-based isobaric labeling, by adding an R script to extract corrected reporter ion intensities from the pertinent MaxQuant *modificationSpecificPeptides.txt* file. We used MaxQuant search results from the publicly available 11-plex TMT dataset PXD009477 from ProteomeXchange as proof of principle. Detailed instructions are provided inside the script *mqTMT_to_ppPeptide.R* that is included in the INKA pipeline code.

## Data availability

The mass spectrometry data have been deposited with the ProteomeXchange Consortium via the PRIDE partner repository (www.ebi.ac.uk/pride/archive) and assigned the identifiers PXD006616, PXD008032, PXD012565, and PXD009995 (http://proteomecentral.proteomexchange.org/cgi/GetDataset).

Code: Researchers can analyze their data by the INKA pipeline at http://www.inkascore.org, download the maintained code from that website (subject to updates) or use the current version of the code provided as Code EV1, which includes a description in file "READ ME.txt".

**Expanded View** for this article is available online.

## Acknowledgements
SK-Mel-28 cells from ATCC were kindly provided by Dr. Elisa Giovannetti, AIRC/Start-Up Unit, Pisa University, Italy. HCC827 ER3 cells were kindly provided by Prof. Balazs Halmos, Division of Hematology/Oncology, Columbia University Medical Center, New York, USA. Raw label-free phosphoproteomic data on human melanoma G361 cells after radiomimetic treatment were kindly provided by Dr. A. Bensimon, ETH Zürich, Switzerland. Inge de Reus is acknowledged for experimental assistance. Dr. Frank Koopman is thanked for critically reading the manuscript. VitrOmics Healthcare Services (VHS) and Cancer Center Amsterdam and Netherlands Organisation for Scientific Research (NWO-Middelgroot project number 91116017) are acknowledged for support of the mass spectrometry infrastructure and Surfsara for computing infrastructure (reference e-infra180166). Furthermore, we thank Cancer Center Amsterdam for support of CvA and TLL, Dutch Cancer Society (projects VU2013-6423, VU2013-6020, VU10212/2016, and NKI2014-6813) for support of FR, JCK, TVP, TYSLL, and AAH and VHS for support of RB.

## Author contributions
CRJ designed the experiments. RB, TVP, JCK, and AAH wrote the R script and performed the data analysis. AAH proposed the combination score, performed INKA, permutation and *in silico* drug analyses, developed the web server and the R script. RB, CA, and RRH performed the experiments. LT and AB provided PDX tumors, and VV assisted with the organoid experiments and culture. CRJ, LT, AB, CA, FR, and TYSLL gave input on the analysis. RB, EH, and SRP optimized NetworKIN settings. SRP performed nanoLC-MS/MS, database searching, and prepared figures. HMWV and ML provided clinical expertise. CRJ, RB, TVP, CA, JCK, AAH, FR, ML, SRP, and HMWV wrote the paper.

## Conflict of interest
The authors declare that they have no conflict of interest.

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
