## [Review Process File · Molecular Systems Biology]

INKA, an integrative data analysis pipeline for phosphoproteomic inference of active kinases

Robin Beekhof, Carolien van Alphen, Alex A. Henneman, Jaco C. Knol, Thang V. Pham, Frank Rolfs, Mariette Labots, Evan Henneberry, Tessa Y.S. Le Large, Richard R. de Haas, Sander R. Piersma, Valentina Vurchio, Andrea Bertotti, Livio Trusolino, Henk M.W. Verheul, and Connie R. Jimenez.

Review timeline:

Submission date:	26 th January 2018
Editorial Decision:	2 nd February 2018
Appeal:	6 th February 2018
Editorial Decision:	16 th February 2018
New submission received:	1 st June 2018
Editorial Decision:	10 th July 2018
Revision received:	18 th January 2019
Editorial Decision:	25 th February 2019
Revision received:	15 th March 2019
Accepted:	20 th March 2019

Editor: Maria Polychronidou

Transaction Report:

1st Editorial Decision

2nd February 2018

Thank you for submitting your manuscript entitled "INKA, an integrative data analysis pipeline for phosphoproteomic inference of active phosphokinases" to Molecular Systems Biology.

I have now had the chance to read your study and I regret to inform you that we have decided to not send it out for peer review.

In this study, you present INKA (Integrative Inferred Kinase Activity), an approach for identifying activated kinases in cancer samples by integrating information both on kinase and substrate phosphorylation. We appreciate that you illustrate INKA by analyzing phosphoproteomics data from different cells lines and report that it correctly detects known oncogenic drivers, activated kinases and it also detects the effect of kinase inhibitor treatments in biopsy samples. While we appreciate these proof-of-principle analyses, we feel that the potential of INKA to identify previously unknown activated kinases that are involved in tumor progression and are therapeutically relevant remains to be further demonstrated, while the applications for personalised medicine remain somewhat tentative. As such while we acknowledge that INKA presents certain improvements over existing kinase-centric or substrate-centric approaches, we are not convinced that the study provides the kind of decisive and broadly relevant methodological advance and the level of biological insight that would be required for publication at Molecular Systems Biology.

I am very sorry to have to disappoint you on this occasion, but I hope that this early decision will allow you to decide how to proceed with your manuscript without undue delay.

Appeal

6th February 2018

Thank you for your fast decision.

Unfortunately for us you have decided not to send our manuscript out for review. Actually I did not expect this, since we showed the potential of our unique analysis strategy using our own as well as published data and also in a different setting uncovered ground truth biology.

The preprint of our paper (<https://www.biorxiv.org/content/early/2018/02/02/259192> <http://proteomicsnews.blogspot.nl/2018/02/inika-find-phosphokinases-in-global.html>) and received a lot of interest in short time with one blog post in which INKA is called a game changer.

Now we are applying INKA successfully in all our current phosphoproteomics projects in different tumor types.

I truly believe our work will be very important for both the cell signaling field and the cancer research community

Is there any chance you could give us a chance and at least send the paper out for review?

Thanks for your consideration and best wishes

2nd Editorial Decision

16th February 2018

Thank you for your message asking us to reconsider our decision on your manuscript MSB-18-8250. I have now discussed your manuscript once again with the Chief Editor and we also have considered the points raised in your letter. As I will explain below, we think that unfortunately there would not be sufficiently compelling reasons to reconsider our decision.

In this study, you propose an approach termed INKA (Integrative Inferred Kinase Activity), for identifying activated kinases in cancer samples. INKA integrates both kinase-centric and substrate-centric evidence to infer kinase activity, in contrast to existing approaches that focus on either the phosphorylation of the kinases themselves or on the phosphorylation of substrates. We appreciate that you illustrate INKA by analyzing phosphoproteomics data from different cells lines with known driver kinases and report that it correctly detects these known oncogenic drivers, and in the case of some drivers it performs better compared to kinase-centric or substrate-centric methods. We also acknowledge that you analyze pY-phosphoproteomic data from patient tumor biopsies collected before and after erlotinib treatment and report that INKA correctly infers the effect of the treatment on EGFR activity. While these proof of principle analyses suggest that the approach might be potentially relevant, we feel that the study remains somewhat preliminary in absence of follow-up analyses showing that the kinases prioritized by INKA are indeed relevant drug targets for single or combinatorial treatment or demonstrating the predictive value of the approach in the context of analyzing drug resistance or personalized responses of different tumors to drugs.

Overall, while we recognize that the topic of the study is relevant, we are still not convinced that the study provides the kind of demonstrated methodological advance with a clear potential to reveal new biological insights that would be required for publication at Molecular Systems Biology. I apologize for not being able to bring better news on this occasion but I hope that the comments above can better explain the reasons behind our decision.

Additional Correspondence with the author:

I have sorted out which data we can free up and add to our INKA manuscript to reveal new insights into oncogenic kinase signaling with functional relevance.

To this end, we can contribute a phosphoproteomics analysis of two different patient-derived xenograft models (human tumor grown in the mouse) of colorectal cancer with functional follow-up in organoids grown from these PDX models. These models were obtained from our Italian collaborator Prof. Livio Trusolino. In organoid cultures we performed dose-response curves of 3 kinase inhibitors (BMS-754807 and Linsitinib against INSR/IGFR1 and afatinib against EGFR/HER2) that were selected based on top ranking kinase activities from the INKA analysis.

Both single treatment with BMS-754807 and Afatinib show a strong inhibition of cellular viability in both models tested as organoids. Linsitinib was not as potent as BMS which may be explained by the higher number of targets and off-targets of BMS in the top 10 INKA kinases of each organoid tested. We also tested the BMS drug and afatinib combined. This combination was not additive probably because blockade of either signalling module (that targeted by BMS and that targeted

by afatinib) can impact the other, and vice versa, due to extensive cross-talks among the involved kinases.

Editor's reply:

Thank you for the follow up on this. I think that these experiments sound potentially interesting. If you extend the manuscript by adding these analyses we will send it out for peer review.

New Submission

1st June 2018

Through the *Molecular Systems Biology* website we submit our manuscript entitled "**INKA, an integrative data analysis pipeline for phosphoproteomic inference of active phosphokinases**".

This is an extended version of the manuscript that we submitted last year (MSB-18-8250R-Q), that now includes data underlining the ability of INKA analysis to reveal new insights into oncogenic kinase signaling and potential drug targets, some of which were functionally tested (see also my email of March 19). As per your advice of March 21 we submit this paper again.

Tools for phosphoproteomic inference of active kinases are in their infancy. The submitted paper describes a method to calculate an Integrative Inferred Kinase Activity (INKA) score to identify highly active kinases in a single biological sample. It uses label-free quantification data on the phosphorylation of kinases, their activation segments, and their substrates as deduced from either established or predicted kinase-substrate relationships.

As a proof of concept, we show for multiple cancer cell lines that INKA scoring identifies top (two) candidate kinases that include known driver kinases, as well as relevant quantitative changes upon perturbation. To illustrate the feasibility of using INKA in a more clinical setting, we also analyze pre- and on-treatment tumor needle biopsies of two patients receiving erlotinib treatment in a phase I clinical study. Finally, in an extension of the original manuscript, we present an INKA analysis of patient-derived xenograft tumors, with functional follow-up showing that kinase inhibitors matched to INKA profiles cause strongly reduced growth in organoid cultures. Via a webserver (www.inkascore.org) we will make the analysis available for the community.

The application of INKA scoring need not be limited to identification of therapeutic targets in a cancer context, but may serve addressing more general biological questions (e.g., DNA damage response signaling, Fig. 5), rendering the paper relevant for a broad readership. We therefore hope that it is suitable for publication in *Molecular Systems Biology*.

3rd Editorial Decision

10th July 2018

Thank you again for submitting your work to *Molecular Systems Biology*. We have now heard back from the three referees who agreed to evaluate your study. As you will see below, the reviewers raise a number of concerns, which unfortunately preclude the publication of the study in its current form.

Overall, the reviewers point out that substantial further analyses are required in order to convincingly demonstrate the superiority of INKA compared to alternative approaches and mention that the biological applications of INKA seem somewhat underdeveloped and lack important controls. However, considering that the reviewers appreciate that INKA sounds potentially interesting and, if well supported, it may be useful for the analysis of phosphoproteomics data, we have decided to offer you a chance to revise the study and address the points raised.

Without repeating all the comments listed below, the most fundamental issues that need to be convincingly addressed are the following:

- Further analyses need to be performed to directly compare INKA with alternative methods/tools and to better support its superiority and advantages.
- The analyses demonstrating the application of INKA to data from cultured cells and patient-derived xenografts need to be expanded to better support the related conclusions. Moreover, appropriate controls should be included. The reviewers provide constructive suggestions related to this point.
- The conclusion that INKA can be applied to single samples needs to be better supported.

- The phosphoproteomics data and the software need to be made available to the reviewers upon submission of the revised version.

- The reviewers mention that the methodology is not described in enough detail. We would ask you to make sure that all information is provided in the main text and is easily accessible to the reader.

All other issues raised by the reviewers would need to be convincingly addressed. As you may already know, our editorial policy allows in principle a single round of major revision. It is therefore essential to provide responses to the reviewers' comments that are as complete as possible.

REFEREE REPORTS.

Reviewer #1:

Beekhof et al. describe in their manuscript a pipeline that integrates four different tools to infer kinase-activity from phospho-proteomics. The tool, INKA, calculates a single score based on these four methods and provides both visualization of the individual results from the tools and ranking based on the combination of the tools. The authors next validate their pipeline using cell line data, patient samples and PDX samples.

The combination of four different methods to measure kinase-activity is interesting as this could, in principle, provide a better score than a single method alone. However, the authors have been unable to show this in the manuscript while a direct comparison with other tools is missing. Furthermore, the performed experiments do not convincingly show that INKA can reveal important druggable targets, since the results are mostly anecdotal and the PDX/organoid experiment lacks essential controls.

The main problems in this manuscript are described in more detail below:

1. The INKA-score calculated on the cell line data, described in figure 2 and 4, highly correlates with the 'Kinome' score. The exact same conclusions for all data can be made by only taking the 'kinome' score into account. The manuscript does not show whether the INKA-score actually improves on the identification of active kinases over existing tools. A direct comparison with the four used tools and other available tools will be needed to show this.
2. The authors conclude the results with the PDX data (Figure 6). The INKA score was calculated on two PDX samples for which phosphoproteomics was performed. Next, the authors tested two drugs on the organoids from these PDX, both of which decreased cell viability.
 - a. First, did the authors check whether the organoid INKA-score was similar to the PDX INKA-score?
 - b. Second, the claim the authors make that "INKA has the ability to guide prioritization of oncogenic drug target candidates" can be made only if the authors show an additional PDX/organoid that does not show the higher INKA scores seen in the PDX and, as expected, will not respond to the tested drug or drugs (single or combination). Without this proper control, no conclusions can be drawn from this experiment.
3. The major part of the results section describes four cell lines, for which it is shown that INKA identifies the known driver pathways and that publicly available drug response data correlates with the INKA-scores. However, this section is overly long, describes hardly any actual results and is more a discussion than a results section.
4. Furthermore, as already discussed for the PDX/organoid data, the authors fail to show the direct link between INKA-score and response. INKA predicts a kinase in a specific cell line to be active and indeed this cell lines responds to the drug targeting this kinase. However, to go beyond an anecdotal example and show that INKA can reveal important drug-targets, the authors need to perform the analysis on multiple cell lines from the same tumor type and show that INKA can predict response to a specific inhibitor. Without these experiments the results are merely anecdotal, similar to the patient samples.
5. The developed software cannot be viewed by the reviewer as the online tool is currently accessible for registered users only. In addition, no statement could be found whether the code to calculate the INKA-score will be made publicly available.

Reviewer #2:

Summary

In this paper, the authors developed a bioinformatic tool (INKA) to infer active kinases from phosphoproteomic data. The novelty of the tool comes from its ability to take into account both kinase centric and substrate centric informations to compute an activity score. The authors then

demonstrate possible uses of their tool in various experimental contexts of cell lines and patients samples. Finally, they validate the predictions of the tool in the context of patient samples by treating organoids derived from corresponding patients with inhibitors of the top scoring kinases predicted by their tool.

The authors claim that INKA allows to pinpoint specific and actionable driver kinases in order to accurately tailor treatments for specific patients, notably in cancer and possibly in other disease contexts. In more details, the authors claim that the tool is especially tailored to handle single sample data and works better with spectral counts rather than and intensity based metric.

The models and methodology were based on a straightforward summation and averaging of spectral counts in different context based on existing knowledge base (mainly differentiating kinases and substrate-centered phosphosite). The normalisation of scores was performed using a two-fold randomisation strategy.

General remarks

In our opinion, most of the authors claim have too many loose ends to be convincing enough. The key concept of the tool which is to combine kinase and substrate is definitely interesting and can very likely improve the predictive power of kinase activity inference methods. However, authors present and execute it in the context of data structures that makes the evaluation of the performances of the tool in comparison to existing methods hard to assess. Multiple approaches toward this goal have already been explored, the works cited by the authors (Rikova et al, 2007; Casado et al, 2013), and beyond (e.g. KEA: kinase enrichment analysis from Ma'ayan lab, or Drake et al PNAS 2012 109, 1643-1648, etc.).

The nature of the advance in this paper is mainly conceptual, as the authors try to integrate mainly two types of approaches to estimate kinase activity. Those two type of approaches are centred on the kinases themselves and on kinase's substrates, respectively. Technically, the methods used by authors are pretty straightforward and the clinical findings are mainly confirmations of previous findings.

This conceptual advance builds up quite naturally and intuitively on existing methods and the concept should definitely be taken into consideration whenever such analysis are performed, assuming that the authors can clearly demonstrate the actual advantage of the conceptual advance. If that is the case, INKA could be relevant to any biologist or bioinformatician that works with phosphoproteomic data, in in-vitro, in-vivo and clinical contexts.

Major points

1. The major criticism comes from the lack of proper benchmark of the methodology against existing methods mentioned above. This becomes critical at several points of the manuscript, and authors should design a proper benchmark to assess the performance of their tool; similar benchmarks exist (e.g. Hernandez-Armenta et al. Bioinf 2017). Related to this authors should provide a more general and thorough revision of existing methods.

1.1. First, they assess the performance of their algorithm by looking at the ranking of the kinases that are known to be altered in specific cell lines (i.e. the higher a kinase is ranked by a method, the better the method). However, this assumption is flawed as the scores they compute simply represent the relative activity of kinases compared to other kinases in the same sample. Just because a kinase has a relatively higher quantity of associated kinase centered and substrate centered phosphopeptides doesn't mean it has necessarily a higher biological importance than other kinase. Because of this, it is not clear if the combination of kinase centric and substrate centric method performs better than each of those methods alone. This could have been partially alleviated by presenting these comparison in a differential setting (comparison of conditions), however the author didn't show such comparison, even though they used the tool in a differential setting.

1.2. Second, even though it was a good idea to use the GDSC data to validate their approach, they only present a few cases where the GDSC is in agreement with the output of the tool, giving an impression of cherry picking. Indeed, there is enough data in the GDSC to always eventually find some positive matches, regardless of the validity of the method used. Actually there are several large-scale phospho screening of cell lines, including one that covers part of GDSC (<https://www.ncbi.nlm.nih.gov/pmc/articles/PMC5583477/>). Such dataset(s) allow a more systematic assessment of INKA.

1.3. The data sets used in the section 'Testing the INKA approach with literature data' are old, why not using newer ones? there are so many recently.

1.4. Since the goal of this tool is to find specific treatable kinases, the validation performed on the organoids should have also been made with healthy organoids and compared. Indeed, just because a treatment reduce the viability of organic tissues doesn't mean it is going to act specifically on cancerous tissue. Authors should confirm the specificity of the driver kinases they target by comparing their results with healthy organoids.

2. The code should be provided for reproducibility and transparency issues. Also the tool would be much more used, and would allow others to build on top of it. There is a webserver but this requires registration.

3. The authors argue that it is critical that tools such as INKA can work with single samples. This is a critical statement, however the author didn't provide enough arguments to be convincing on this matter. Indeed, the interpretation of results obtained in single samples is completely different of results obtained in the context of differential analysis, and even more when biological replicates are available. In a single sample, as said above, scores are only relative to other kinases in the same sample, which is insufficient to conclude about the biological relevance of a kinase. Authors provide more convincing arguments as to why it is necessary for such tools to work in the context of single disease samples (without comparisons with healthy samples). Even in clinical context it is usually possible to obtain both healthy and diseased tissue samples from patients.

Minor points

4. The author claims that the tool works better with spectral counts than with intensity based metrics. The authors should provide data to back up this claim.

5. In the introduction, the authors could be more clear when they explain the difference between kinase and substrate centered approaches.

Reviewer #3:

In the present manuscript, Beekhof et al present INKA, a data analysis pipeline for inference of the activity of protein kinases in human cancer cells. At the heart of the method is the idea to combine the estimated quantities of kinase and substrate phosphorylation into a single score that represents the activation status of said kinase in a cell. Overall, this is a good idea because the approach may be able to focus the long lists of proteins coming out of phosphoproteomic studies of cancer cell lines or tumors down to a group of proteins that often represent drug targets and can therefore potentially lead to clinically actionable results. The authors also claim that the method would enable interpretation of individual samples (patients) which is of paramount importance when it comes to taking treatment decisions on individual patients. In that sense, the study goes beyond the current state of the art in which the analysis of phosphoproteomes usually takes the form of cohorts and clustering of some sort to define patient subgroups. So, overall, the study presents an original idea and the data provided indicates that the method has merit. That said, the authors have to address quite a few issues before the work may eventually be fit for publication in MSB.

- Although the raw MS data has been deposited with PRIDE, the authors do not provide a login for reviewers. Hence, it is impossible to judge the quality of the underlying data. I could not find a cover letter in the submission where this information may have been supplied. Also, the MaxQuant output files are not supplied and the suppl table listing all the p-peptides is missing important information including which p-site was actually assigned. The authors have to provide much more information in order to enable reviewers to take a close look at the data. Perhaps all this is in the PRIDE submission but it is locked up, so one cannot get to it. In addition, the methods section lacks a lot of important information, notably if/how data was normalised (important for e.g. the drug treatment data).

- The authors claim that the method can be used on single samples. Given the way they calculate their score, at least one half of it (the kinase-centric one) is essentially trying to absolutely quantify the phosphorylated portion of kinases so that they can be assembled into a ranked list. Here, they fail to normalize their kinase-centric score to the length of the protein (analogous to iBAQ). Their score will rank bigger kinases with a higher propensity to be phosphorylated higher. The same is true for their substrate-centric analysis (second half of the INKA score) in case they do not restrict the analysis to the p-site for which they know the upstream kinase (I guess this is what is being done). Kinases with bigger substrates would rank higher in this part of their score. I suspect this will

strongly influence their result and skew INKA scores of bigger kinases with on average bigger substrates. The authors should clarify this and provide an analysis that compares the normalized and current way of calculating INKA.

- The kinase activation loop phosphopeptides were already used in the calculation of the first part of their score. It would seem wrong to use it twice. It would be instructive to see whether the INKA score would drastically change would one leave out the second part of their kinase-centric analysis.

- Moreover, their score will scale with measurement depth. This means that if they happen to measure a tumour sample on one day on a clean instrument and get many phosphopeptides for say ABL1 and its substrates (high INKA score), the same run on a poorly performing instrument will result in a lower INKA score. Dealing with such technical things may be easy in a well controlled lab environment but as the patient data in Figure 4 shows, the range of INKA scores can be quite different from sample to sample. Related, the authors should discuss how INKA scores are (not) comparable between laboratories, which is suboptimal for clinical decision making.

- All of their scores are based on spectral counts, which are - as they claim - less sensitive to outlier phosphopeptides with very high abundance. But maybe it's these outliers that are actually interesting! Related: because the data is available, the authors should check if INKA can be improved when using the peptide intensity provided by MaxQuant. Spectral counts are semi-quantitative at best and only work (well) for high spectral counts. Obviously, using LC-based quantification only works if the LC and MS parameters have been matched such that enough data points were collected across the LC peak. I could not get to this information. Related, the paper is in need for discussing the shortcomings of SC. This particularly shows in the analysis of the patient data in Figure 4 where INKA scores are quite low and vary between samples quite a lot. I understand that INKA provides a ranked list of kinases to consider but it is a stretch to interpret the EGFR finding in Figure 4C and 4D given the low INKA scores (I guess few p-peptides). Looking at the kinases on the list of which quite a few are ranked higher than EGFR, a SRC family inhibitor such as Dasatinib would have made more sense than Erlotinib.

- In general, the quality of the data is sometimes hard to judge. Again, as interesting as Figure 4 is, we do not know how significant the observed changes are. In the EGFR overexpression system, the data is probably fairly robust (judging from high INKA scores implying high spectrum counts) but this is much less clear in the patient data. This is particularly important as the drug treatment of the cell lines always led to reduced (or unchanged) INKA scores but in the patient data there are cases with increased INKA scores. These are hard to interpret in the absence of replicates/error bars/confidence levels. For the same reason, it is hard to interpret the extent of INKA reduction for EGFR because it is unclear how many spectral counts underpin the INKA calculation

- They calculate p-values for their scores based on empirical null distributions from permutations. And then they claim that the correlation of low p-values with high INKA-scores "underscores the relevance of the INKA score". However, this is a self-fulfilling prophecy. The higher any of the individual scores contributing to INKA are (kinase-centric, activation loop, PSP, NWK), the less likely it will be to replace it with an even higher score from the permutations. This has nothing to do with the relevance of the INKA score.

- They then go ahead and say the INKA score can be used in differential analyses and that it predicts drug sensitivity. However, they fail to actually correlate the INKA score to the GDSC data or do something like ElasticNet regression. Would an approach like the ElasticNet select e.g. ABL1 based on INKA score when looking at, say, imatinib? I appreciate that the number of cell lines available in their data set may preclude such an analysis. However, at least for some of the cell lines included in the GDSC panel, there is public p-proteome data.

- The authors claim that it is a good idea to calculate the INKA score only based on kinase phosphosites when there are no substrates known, but don't calculate INKA score when a kinase was not detected but their substrates were present, which makes sense. They say: "For kinases inferred through PhosphoSitePlus/NetworkKIN but not observed by MS, the reciprocal analysis is not performed, as kinases display overlapping substrate specificities precluding unequivocal assignment of a substrate to a specific kinase." However, they then go ahead and interpret their data on SK-MEL-28 (BRAfV600E) in the light of downstream substrates of BRAF and claim that high INKA scores of downstream kinases means the upstream kinase must have been active. Maybe they should do regular (e.g. IMAC or TiO₂) phosphopeptide enrichment and not pY-IPs and see whether BRAF ends up with the top INKA score.

- When looking at HCC827-ER3, they say that these cells are highly sensitive to EGFR inhibitors and refer to Supplementary Table 4. There, -1.99 and even -1.17 (z-score) is apparently highly sensitive. Later, they focus on ALK and alectinib and say -1.92 (again z-score) is not sensitive. Since the z-score makes sensitivities somewhat comparable between drugs, they interpret the same data differently depending on their expectations.

- Figure 6 and Supplementary Figure 9 are identical. I therefore cannot judge Supplementary Figure 9. However, in Figure 6D, they claim that treating PDX with a selection of drugs targeting kinases with high INKA scores highlights the INKA score as a potential tool for personalized medicine.

- However, they fail to check whether these drugs would have also killed PDX in which other kinases scored high using INKA! Maybe these compounds just generally work well in their PDX model.

- There is no direct comparison to e.g. KSEA.

- I could not use their website, since it requires a login.

- Use of the word phosphokinases in the title is weird.
- Manuscript suffers from poor use of the English language in some paragraphs.

1st Revision - authors' response

18th January 2019

We thank the reviewers for their time and thoughtful comments. Below we explain in detail how we addressed the points raised by the reviewers.

Reviewer #1:

Beekhof et al. describe in their manuscript a pipeline that integrates four different tools to infer kinase-activity from phospho-proteomics. The tool, INKA, calculates a single score based on these four methods and provides both visualization of the individual results from the tools and ranking based on the combination of the tools. The authors next validate their pipeline using cell line data, patient samples and PDX samples.

The combination of four different methods to measure kinase-activity is interesting as this could, in principle, provide a better score than a single method alone. However, the authors have been unable to show this in the manuscript while a direct comparison with other tools is missing. Furthermore, the performed experiments do not convincingly show that INKA can reveal important druggable targets, since the results are mostly anecdotal and the PDX/organoid experiment lacks essential controls.

The main problems in this manuscript are described in more detail below:

1. The INKA-score calculated on the cell line data, described in figure 2 and 4, highly correlates with the 'Kinome' score. The exact same conclusions for all data can be made by only taking the 'kinome' score into account. The manuscript does not show whether the INKA-score actually improves on the identification of active kinases over existing tools. A direct comparison with the four used tools and other available tools will be needed to show this.

ANSWER

We thank the reviewer for the encouraging words and appreciation that INKA can provide a better score than a single method alone.

For all 7 cancer cell line examples displayed in figures 2 and 4 with known genomically aberrant oncogenic kinases, we not only show the INKA ranking, but also show the ranking of the 4 INKA components: phosphorylated kinases ('Kinome' component), phosphorylated kinase activation loops ('Activation Loop' component), kinase prediction for observed phosphoproteins, with prediction based on the PhosphoSitePlus kinase-substrate relationships database ('PhosphoSitePlus' component), and kinase predictions for observed phosphoproteins based on the NetworKIN algorithm that leverages phosphorylation motif and protein-protein interaction knowledge ('NetworKIN' component).

Based on these separate analyses represented in parallel, we observe that each INKA component yields overlapping but also different information. Of the 4 components, the Kinome-based ranking is clearly a very important layer of information that performs very well (results section pg 9 ln 150-152). This is an important finding since most other approaches for inference of kinase activity make use of substrate-based analyses only, and an avenue that is more prone to false positive results. The important layer of information provided by the kinome is in line with observations by others that kinase hyperphosphorylation is associated with increased kinase activity (e.g. Rikova et al., Cell 131, 1190, 2007). Yet, based on the Kinome component alone, one cannot deduce with confidence whether a kinase was active. Therefore, the observation that the more stringent INKA approach consistently assigns a high rank to expected driver kinases in all cancer cell lines analyzed is important. It also provides a network view of kinase-substrate relations for top 20 kinases, which sheds more light on interrelated signaling biology and helps prioritization. Moreover, in the resubmission we now also compare INKA and its 4 components to one other published single sample tool, KARP (new supplementary figure Appendix Fig S7) that makes use of substrate data from PhosphoSitePlus. This comparison shows the superiority of INKA over KARP for oncogene-driven cell lines in ranking the driver kinase(s) (high).

Added text (pg 10 ln 166-170)

The INKA score was compared to KARP (Wilkes et al, 2017), another kinase activity ranking tool

that can be used on single samples. KARP kinase activity ranking is based on substrate phosphorylation analysis in combination with kinase-substrate relations. For the four oncogene-driven cell lines, INKA outperformed KARP in assigning high ranks to the known drivers (Appendix Fig S7).

2. The authors conclude the results with the PDX data (Figure 6). The INKA score was calculated on two PDX samples for which phosphoproteomics was performed. Next, the authors tested two drugs on the organoids from these PDX, both of which decreased cell viability.
- a. First, did the authors check whether the organoid INKA-score was similar to the PDX INKA-score?

ANSWER

The reviewer raises an important point. We now performed phosphoproteomics and INKA analysis on organoids, indicating consistency in INKA profiles in models CRC0177 and CRC0254. In CRC0177, INSR scores high in both PDX and organoid; the same can be observed for other kinase activities such as EGFR and ERBB2, together explaining the sensitivity to afatinib and the stronger effect of the combination treatment with BMS. The same holds for CRC0254: EPHB3, EGFR, INSR, SRC are high ranking in both PDX and organoid, and sensitivity to BMS, afatinib and BMS + afatinib can be rationally explained on the basis of our results.

Added text (pg 14 ln 315-317)

Importantly, INKA analysis of PDX-derived organoids showed overall consistency of target activity in both models CRC0177 and CRC0254 (Figure 7C).

- b. Second, the claim the authors make that "INKA has the ability to guide prioritization of oncogenic drug target candidates" can be made only if the authors show an additional PDX/organoid that does not show the higher INKA scores seen in the PDX and, as expected, will not respond to the tested drug or drugs (single or combination). Without this proper control, no conclusions can be drawn from this experiment.

Answer

To address the concern of the reviewer regarding negative control, we selected a low ranking INKA kinase for targeting, i.e., ABL that ranks low in both CRC0177 and CRC0254 PDX INKA profiles, and low ranking ABL activity is also preserved in the corresponding organoids. Organoid treatment with imatinib, as expected from the low ranking, showed negligible inhibition of organoid viability (fig. 6 and appendix fig. 9). Only at a high imatinib concentration (10 μ M), cell-viability was affected in both organoid models.

Added text (pg14 ln326-331)

To explore whether a kinase with a low INKA score does not show a response to the corresponding drug, we selected ABL that ranked low in both PDXs and organoids of CRC0177 and CRC0254. Indeed, organoid treatment with the ABL inhibitor imatinib yielded negligible inhibition (IC50 imatinib = 4 or 6 μ M for CRC0177 and CRC0254, respectively) (Fig. 7D) while the positive control (CML cell line K562) worked (supplementary figure S11E), underscoring the value of INKA ranking for drug response prediction.

3. The major part of the results section describes four cell lines, for which it is shown that INKA identifies the known driver pathways and that publicly available drug response data correlates with the INKA-scores. However, this section is overly long, describes hardly any actual results and is more a discussion than a results section.

Answer

We shortened the paragraphs on the four use case cell lines as much as possible.

4. Furthermore, as already discussed for the PDX/organoid data, the authors fail to show the direct link between INKA-score and response. INKA predicts a kinase in a specific cell line to be active and indeed this cell lines responds to the drug targeting this kinase. However, to go beyond an anecdotal example and show that INKA can reveal important drug-targets, the authors need to perform the analysis on multiple cell lines from the same tumor type and show that INKA can

predict response to a specific inhibitor. Without these experiments the results are merely anecdotal, similar to the patient samples.

Answer

The reviewer raises an important point.

Only very few label-free phosphoproteomics datasets are available in the public domain and they are limited in size. Nevertheless, we now include a public PDAC pTyr dataset composed of 11 cell lines with drug efficacy data (Humphrey et al. Resolution of Novel Pancreatic Ductal Adenocarcinoma Subtypes by Global Phosphotyrosine Profiling. Mol Cell Proteomics. 2016 2671-85.) and a public CRC TiOx dataset of 8 cell lines. We have now added the analysis of these panels of cancer cell lines, in addition to the five cell lines we analysed ourselves, for which drug data is available. In the new results section ‘Comparing INKA to its components, to KARP, and correlation with drug efficacy’ we describe the analysis in which we compare INKA, its components and KARP (wilkes et al MCP 2017), a single-sample substrate-based kinase activity inference method. We introduce the kinase impact score and apply this algorithm to the kinase ranking methods above. The kinase impact score correlates kinase activity ranked lists with cell line IC50 values of drugs in GDSC. We analyzed pTyr IP data and TiO₂ global phosphoproteomics data in two separate analyses with in total 20 distinct cell lines and a statistical analysis of significance. The kinase impact score is globally higher for INKA than for KARP, or for the four INKA components, indicating that INKA is superior in ranking kinase activities in the context of drug efficacy.

We added the new Fig 6 where the results of this analysis are shown, as well as a results section (Pg 14-15 Ln 270-298), and expanded the materials and methods section (Pg 38-40 Ln 810-849) accordingly.

5. The developed software cannot be viewed by the reviewer as the online tool is currently accessible for registered users only. In addition, no statement could be found whether the code to calculate the INKA-score will be made publicly available.

Answer

We now compiled an R version of the script and added a sentence at the end of the introduction, in the results section, and before the references, that the R script is available for download at www.inkascore.org

Added text (Pg5 Ln67-68)

INKA is available through a web server at www.INKAscore.org and as R script

Added text (Pg8 Ln133-135)

The INKA analysis pipeline is available both as a web service at <http://www.inkascore.org> and as a standalone R script.

Added text (Pg42 Ln903-905)

Code Availability

Researchers can analyze their data by the INKA pipeline or download the R-code from www.inkascore.org.

Reviewer #2:

Summary

In this paper, the authors developed a bioinformatic tool (INKA) to infer active kinases from phosphoproteomic data. The novelty of the tool comes from its ability to take into account both kinase centric and substrate centric informations to compute an activity score. The authors then demonstrate possible uses of their tool in various experimental contexts of cell lines and patients samples. Finally, they validate the predictions of the tool in the context of patient samples by treating organoids derived from corresponding patients with inhibitors of the top scoring kinases predicted by their tool.

The authors claim that INKA allows to pinpoint specific and actionable driver kinases in order to accurately tailor treatments for specific patients, notably in cancer and possibly in other disease

contexts. In more details, the authors claim that the tool is especially tailored to handle single sample data and works better with spectral counts rather than and intensity based metric.

The models and methodology were based on a straightforward summation and averaging of spectral counts in different context based on existing knowledge base (mainly differentiating kinases and substrate-centered phosphosite). The normalisation of scores was performed using a two-fold randomisation strategy.

General remarks

In our opinion, most of the authors claim have too many loose ends to be convincing enough. The key concept of the tool which is to combine kinase and substrate is definitely interesting and can very likely improve the predictive power of kinase activity inference methods. However, authors present and execute it in the context of data structures that makes the evaluation of the performances of the tool in comparison to existing methods hard to assess. Multiple approaches toward this goal have already been explored, the works cited by the authors (Rikova et al, 2007; Casado et al, 2013), and beyond (e.g. KEA: kinase enrichment analysis from Ma'ayan lab, or Drake et al PNAS 2012 109, 1643-1648, etc.).

The nature of the advance in this paper is mainly conceptual, as the authors try to integrate mainly two types of approaches to estimate kinase activity. Those two type of approaches are centred on the kinases themselves and on kinase's substrates, respectively. Technically, the methods used by authors are pretty straightforward and the clinical findings are mainly confirmations of previous findings.

This conceptual advance builds up quite naturally and intuitively on existing methods and the concept should definitely be taken into consideration whenever such analysis are performed, assuming that the authors can clearly demonstrate the actual advantage of the conceptual advance. If that is the case, INKA could be relevant to any biologist or bioinformatician that works with phosphoproteomic data, in in-vitro, in-vivo and clinical contexts.

Answer

We thank the reviewers for their positive words and thoughtful comments.

Major points

1. The major criticism comes from the lack of proper benchmark of the methodology against existing methods mentioned above. This becomes critical at several points of the manuscript, and authors should design a proper benchmark to assess the performance of their tool; similar benchmarks exist (e.g. Hernandez-Armenta et al. Bioinf 2017). Related to this authors should provide a more general and thorough revision of existing methods.

Answer

We agree with the reviewer that the INKA pipeline is a conceptual advance. The article that the reviewer refers to is an extensive benchmarking of existing, previously developed analysis tools to pinpoint differential kinase activity on a gold standard dataset of fold changes. However, this dataset as such cannot be used as input for the INKA pipeline, as that was not designed to perform group comparisons (fold-change values). We would like to emphasize that single sample workflows do not exist except for one method called KARP made by the Cutillas team that we estimate is conceptually similar to the PSP-based INKA component (Wilkes et al., Kinase activity ranking using phosphoproteomics data (KARP) quantifies the contribution of protein kinases to the regulation of cell viability MCP 2017).

Therefore, now we not only benchmark INKA against its individual components (Figure 2) but also compare INKA to KARP directly. To this end, we implemented and ran KARP on the 4 cancer cell lines of Figure 2. The comparison of INKA vs KARP (Appendix Fig S7) shows the superiority of INKA over KARP for oncogene-driven cell lines in (high) ranking the driver kinase(s). The text has been modified

Added text (pg 10 ln 166-170)

The INKA score was compared to KARP (Wilkes et al, 2017), another kinase activity ranking tool

that can be used on single samples. KARP kinase activity ranking is based on substrate phosphorylation analysis in combination with kinase-substrate relations. For the four oncogene-driven cell lines, INKA outperformed KARP in assigning high ranks to the known drivers (Appendix Fig S7).

Furthermore, we have now added an analysis of a panel of cancer cell lines for which drug data is available. In the new results section 'Comparing INKA to its components, and to KARP, and correlation with drug efficacy' we describe an analysis to compare INKA, its components and KARP (Wilkes et al MCP 2017), a single-sample substrate-based kinase activity inference method. We introduce the kinase impact score and apply this algorithm to the kinase ranking methods above. The kinase impact score correlates kinase activity ranked lists with cell line IC50 values of drugs in GDSC. We used publicly available label-free pTyr IP data as well as TiO₂ global phosphoproteomics data, in addition to cell lines described in this manuscript, in two separate analyses with in total 20 distinct cell lines and a statistical analysis of significance. The kinase impact score is globally higher for INKA than for KARP, or for the four INKA components, indicating that INKA is superior in ranking kinase activities in the context of drug efficacy. We added the new Fig 6 where the results of this analysis are shown, as well as a results section (Pg 14-15 Ln 270-298), and expanded the materials and methods section (Pg 38-40 Ln 810-849) accordingly.

1.1. First, they assess the performance of their algorithm by looking at the ranking of the kinases that are known to be altered in specific cell lines (i.e. the higher a kinase is ranked by a method, the better the method). However, this assumption is flawed as the scores they compute simply represent the relative activity of kinases compared to other kinases in the same sample. Just because a kinase has a relatively higher quantity of associated kinase centered and substrate centered phosphopeptides doesn't mean it has necessarily a higher biological importance than other kinase.

Answer

Indeed, INKA ranks kinases on their relative kinase activities based on direct kinase phosphorylation and kinase substrate phosphorylation. Biological importance of a kinase is network and cell context dependent and need not be correlated with relative activity ranking. However, using ranking by kinase phosphorylation alone (kinase centered) Rikova et al. convincingly showed that high-ranking kinases are biologically important (Cell 131, 1190, 2006) and this ranking allows identification of novel oncogenic drivers (EML4-ALK fusion, fig 2D). Especially amplification-driven oncogenic kinases or constitutively active kinase(-fusions) rank high by INKA (e.g. EGFRvIII in GBM fig 5A and BCR-ABL fusion in CML fig 2A and EML4-ALK fusion in Fig 2D). The kinases ranking high in the INKA analyses of these cell lines are the biologically important drivers in these examples. Moreover, our drug experiment in organoids underscores the functional relevance of high ranking kinase activities (INSR, EGFR, ERBB2) as compared to low ranking (ABL). We mention this in the results section:

Added text (pg 9 Ln 158-160)

Altogether these results show that amplification-driven oncogenic kinases or constitutively active kinase(-fusions) rank high by INKA, in line with previous findings (Rikova et al, 2007; Guo et al, 2008).

Because of this, it is not clear if the combination of kinase centric and substrate centric method performs better than each of those methods alone. This could have been partially alleviated by presenting these comparison in a differential setting (comparison of conditions), however the author didn't show such comparison, even though they used the tool in a differential setting.

Answer

We thank the reviewer for the suggestion. We added a supplementary figure Appendix Fig S9 showing the INKA components of the differential INKA data shown in figure 5. Text is added in the results section:

Added text (pg 13-14 Ln 247-257)

In Appendix Fig S9, differential analyses at the individual INKA component levels (kinome, activation loop, PSP, NWK) can be found. For the comparisons of U87 wild-type versus EGFR-mutant cells (Appendix Fig S9A) and untreated mutant cells versus erlotinib-treated mutant cells

(Appendix Fig S9B) INKA components are similar. All four components indicate lower EGFR phosphorylation in wild-type relative to mutant cells and erlotinib-treated relative to untreated cells, respectively, while, e.g., MET phosphorylation is not affected by erlotinib treatment. For low-level input samples from patients (Appendix Fig S9C,D), the combined INKA score shows the more robust EGFR response, compared to the individual INKA components. Taken together, combination of the four INKA components into a single score averages out noise and results in a more robust kinase activity ranking than each individual component by itself, and can also be applied in a differential setting.

1.2. Second, even though it was a good idea to use the GDSC data to validate their approach, they only present a few cases where the GDSC is in agreement with the output of the tool, giving an impression of cherry picking. Indeed, there is enough data in the GDSC to always eventually find some positive matches, regardless of the validity of the method used. Actually there are several large-scale phospho screening of cell lines, including one that covers part of GDSC (<https://www.ncbi.nlm.nih.gov/pmc/articles/PMC5583477/>). Such dataset(s) allow a more systematic assessment of INKA.

Answer

We agree with the reviewer that it would be valuable to extrapolate the approach to phosphoproteomics data of a panel of cancer cell lines with drug sensitivity data.

The reviewer refers to a study in which TMT labeling was used (Roumeliotis, Cell Rep, 2017). However, our pipeline was designed to handle label-free phosphoproteomics data. Moreover, when we tried to adapt the CRC data to the INKA pipeline, we noticed strong batch effects of the reference sample (SW480) precluding INKA-based phosphoproteomics data analysis, unfortunately. Here we repeat our answer above:

Only very few label-free phosphoproteomics datasets are available in the public domain and they are limited in size. Nevertheless, we now include a public PDAC pTyr dataset composed of 11 cell lines with drug efficacy data (Humphrey et al. Resolution of Novel Pancreatic Ductal Adenocarcinoma Subtypes by Global Phosphotyrosine Profiling. Mol Cell Proteomics. 2016 2671-85.) and a public CRC TiO₂ dataset of 4 cell lines. We have now added the analysis of these panels of cancer cell lines, in addition to the five cell lines we analysed ourselves, for which drug data is available. In the new results section 'Comparing INKA to its components, to KARP, and correlation with drug efficacy' we describe the analysis in which we compare INKA, its components and KARP (Wilkes et al MCP 2017), a single-sample substrate-based kinase activity inference method. We introduce the kinase impact score and apply this algorithm to the kinase ranking methods above. The kinase impact score correlates kinase activity ranked lists with cell line IC₅₀ values of drugs in GDSC. We analyzed pTyr IP data and TiO₂ global phosphoproteomics data in two separate analyses with in total 20 distinct cell lines and a statistical analysis of significance. The kinase impact score is globally higher for INKA than for KARP, or for the four INKA components, indicating that INKA is superior in ranking kinase activities in the context of drug efficacy.

We added the new Fig 6 where the results of this analysis are shown, as well as a results section (Pg 14-15 Ln 270-298), and expanded the materials and methods section (Pg 38-40 Ln 810-849) accordingly.

1.3. The data sets used in the section 'Testing the INKA approach with literature data' are old, why not using newer ones? there are so many recently.

Answer

We extensively looked for label-free phosphotyrosine IP-based phosphoproteomics data as we think this datatype is most relevant for the large number of clinically used tyrosine kinase inhibitors. Yet that kind of data is quite rare. We found one additional dataset of pancreatic cancer cell lines (Humphrey Mol Cell Proteomics 15, 2671, 2016) that we now also analyze in the same way as outlined under 1.2 above. See further under 1.2.

1.4. Since the goal of this tool is to find specific treatable kinases, the validation performed on the organoids should have also been made with healthy organoids and compared. Indeed, just because a treatment reduce the viability of organic tissues doesn't mean it is going to act specifically on cancerous tissue. Authors should confirm the specificity of the driver kinases they target by comparing their results with healthy organoids.

Answer

It is well established that oncogenes have unusual high activity that is not observed in normal tissues. Furthermore, every drug has its own toxicity profile that can occur in a range of tissues. Normal colon tissues were not collected. Therefore normal organoids were not made and tested.

2. The code should be provided for reproducibility and transparency issues. Also the tool would be much more used, and would allow others to build on top of it. There is a webserver but this requires registration.

Answer

We provide both a webserver to do the INKA analysis and now we also developed an R version of the script that we provide the downloadable R-code

Added text (Pg5 Ln67-68)

INKA is available through a web server at www.INKAscore.org and as R script

Added text (Pg8 Ln133-135)

The INKA analysis pipeline is available both as a web service at <http://www.inkascore.org> and as a standalone R script.

Added text (Pg42 Ln903-905)**Code Availability**

Researchers can analyze their data by the INKA pipeline or download the R-code from www.inkascore.org.

3. The authors argue that it is critical that tools such as INKA can work with single samples. This is a critical statement, however the author didn't provide enough arguments to be convincing on this matter. Indeed, the interpretation of results obtained in single samples is completely different of results obtained in the context of differential analysis, and even more when biological replicates are available. In a single sample, as said above, scores are only relative to other kinases in the same sample, which is insufficient to conclude about the biological relevance of a kinase. Authors provide more convincing arguments as to why it is necessary for such tools to work in the context of single disease samples (without comparisons with healthy samples). Even in clinical context it is usually possible to obtain both healthy and diseased tissue samples from patients.

Answer

To speak with the words of reviewer 3: "the method would enable interpretation of individual samples (patients) which is of paramount importance when it comes to taking treatment decisions on individual patients." We added text to provide more background to the need for a single sample pipeline.

Added text (pg 4 ln 41-43)

This is pivotal in a clinical setting, where one wishes to prioritize actionable kinases for treatment selection for individual patients.

Minor points

4. The author claims that the tool works better with spectral counts than with intensity based metrics. The authors should provide data to back up this claim.

Answer

The INKA analysis using the intensity data for the cell lines in Fig 2 is now provided in Appendix Fig S12. Comparison of the analyses shows that count-based INKA outperforms intensity-based INKA in the task of ranking oncogenic driver kinases high in the 4 cancer cell lines in Fig 2. In 2/4 cases counts and intensities showed the same (top-3) driver kinase rank and in 2/4 cases INKA ranked the driver kinase substantially higher than the intensity based analysis. Additionally, we plotted individual biological replicate analyses that show highly similar ranking and INKA scores across replicates.

Added text (Pg40 ln 851-857)**INKA: spectral counts versus Intensity**

Although INKA analysis can be performed with intensity-based quantification, we favor spectral count-based quantification as it is less sensitive to peptides with outlier intensities and is more robust for the analysis of aggregated data for multiple peptides, some of which may exhibit dominantly high intensities. For Q Exactive data, spectral counting outperformed intensity-based quantification for INKA-based kinase ranking of known drivers (Appendix Fig S12), yet for the low level LTQ-FTMS data the intensity data worked better (Fig 5E,F).

5. In the introduction, the authors could be more clear when they explain the difference between kinase and substrate centered approaches.

Answer

We added a clarifying phrase in the introduction, Pg 4-5 ln 44-53 and the text now reads : Different kinase ranking approaches have been described previously. Rikova and colleagues sorted kinases on the basis of the sum of the spectral counts (an MS correlate of abundance) for all phosphopeptides attributed to a given kinase, and identified known and novel oncogenic kinases in lung cancer (Rikova et al, 2007). This type of analysis can be performed in individual samples, but is limited by a focus on phosphorylation of the kinase itself, rather than the (usually extensive) set of its substrates. Instead, several substrate-centric approaches, focusing on phosphopeptides derived from kinase targets, also exist, including KSEA (Casado et al, 2013; Terfve et al, 2015; Wilkes et al, 2015), pCHIPS (Drake et al, 2016), and IKAP (Mischnik et al, 2016). The only single-sample implementation of substrate-centric kinase-activity analysis is KARP and has been reported recently (Wilkes et al, 2017).

Reviewer #3:

In the present manuscript, Beekhof et al present INKA, a data analysis pipeline for inference of the activity of protein kinases in human cancer cells. At the heart of the method is the idea to combine the estimated quantities of kinase and substrate phosphorylation into a single score that represents the activation status of said kinase in a cell. Overall, this is a good idea because the approach may be able to focus the long lists of proteins coming out of phosphoproteomic studies of cancer cell lines or tumors down to a group of proteins that often represent drug targets and can therefore potentially lead to clinically actionable results. The authors also claim that the method would enable interpretation of individual samples (patients) which is of paramount importance when it comes to taking treatment decisions on individual patients. In that sense, the study goes beyond the current state of the art in which the analysis of phosphoproteomes usually takes the form of cohorts and clustering of some sort to define patient subgroups. So, overall, the study presents an original idea and the data provided indicates that the method has merit. That said, the authors have to address quite a few issues before the work may eventually be fit for publication in MSB.

- Although the raw MS data has been deposited with PRIDE, the authors do not provide a login for reviewers. Hence, it is impossible to judge the quality of the underlying data. I could not find a cover letter in the submission where this information may have been supplied.

Answer

We are a bit puzzled since accession numbers were provided in the cover letter, which stated "I have attached login details that reviewers can use to access data sets and utilize the INKA webtool". The attachment was referred to at the bottom of the cover letter: "Attachment: Sheet with login details for data sets deposited with ProteomeXchange and for access to the INKAscore web-based tool."

Login INKA details: "Username: inkareviewer, Password: oncoproteomics"

Also, the MaxQuant output files are not supplied and the suppl table listing all the p-peptides is missing important information including which p-site was actually assigned. The authors have to provide much more information in order to enable reviewers to take a close look at the data. Perhaps all this is in the PRIDE submission but it is locked up, so one cannot get to it. In addition, the methods section lacks a lot of important information, notably if/how data was normalised (important for e.g. the drug treatment data).

Answer

Indeed these tables were in the PRIDE submissions for which the accession numbers were in the cover letter. Excerpts of the MaxQuant “modificationSpecificPeptides.txt” and “Phospho (STY)Sites.txt” tables were provided in Dataset EV1 (referred to in the text on pg 6 ln 78). The full MaxQuant output tables are deposited in proteomexchange (search). For drug treatment data in Fig 5 indeed raw INKA scores (no normalization) were used for all comparisons.

- The authors claim that the method can be used on single samples. Given the way they calculate their score, at least one half of it (the kinase-centric one) is essentially trying to absolutely quantify the phosphorylated portion of kinases so that they can be assembled into a ranked list. Here, they fail to normalize their kinase-centric score to the length of the protein (analogous to iBAQ). Their score will rank bigger kinases with a higher propensity to be phosphorylated higher. The same is true for their substrate-centric analysis (second half of the INKA score) in case they do not restrict the analysis to the p-site for which they know the upstream kinase (I guess this is what is being done). Kinases with bigger substrates would rank higher in this part of their score. I suspect this will strongly influence their result and skew INKA scores of bigger kinases with on average bigger substrates. The authors should clarify this and provide an analysis that compares the normalized and current way of calculating INKA.

Answer

We thank the reviewer for the iBAQ suggestion and investigated the effect of extending INKA with the iBAQ protein quantification normalization method (Global quantification of mammalian gene expression control, Björn Schwanhäusser, Dorothea Busse, Na Li, Gunnar Dittmar, Johannes Schuchhardt, Jana Wolf, Wei Chen & Matthias Selbach, Nature 2011, 337–342). Originally, the method involves division of the total number of spectral counts (or Intensity) attributed to a protein, by the corresponding number of detectable tryptic peptides. Extension of this to the INKA score calculation involved on the substrate side dividing the PSP and NWK spectral counts of each kinase, by the number of kinase-substrate (KS) relations present in the respective KS-networks for the kinase under consideration. On the kinase side of the calculation, kinome counts were divided by the number of kinase peptides that contained an amino acid that could be phosphorylated, and were at least 7 amino acids long. For pTyr IP-experiments at least one tyrosine should be present in the peptide, whereas for TiOx experiments, also peptides containing at least a serine or threonine were taken into account. The activation loop peptide contribution was left unchanged. For the eight samples considered (4 cell lines of Fig 2 and the four U87 conditions of Fig 5A,B), the highest rank for the driver kinases were either equivalent or better for uncorrected INKA score calculations. Therefore, we decided not to include the iBAQ approach into INKA (see Appendix Fig13).

We added text (Pg40-41 Ln859-872) to the materials and methods section:

iBAQ correction for INKA

We explored implementation of an iBAQ procedure (Schwanhäusser et al, 2011) to correct for the number of phosphopeptides per kinase and the number of substrates per kinase. For the substrate side of INKA we divided the PSP and NWK spectral counts of each kinase, by the number of kinase-substrate relations present in the respective kinase-substrate networks for the kinase under consideration. On the kinase side of INKA, kinome counts were divided by the number of kinase peptides that contained an amino acid that could be phosphorylated, and were at least 7 amino acids long. For pTyr IP experiments, at least one tyrosine should be present in the peptide, whereas for TiO₂ experiments, also peptides containing at least a serine or threonine were taken into account. The activation loop peptide contribution was left unchanged. For the eight samples considered (four cell lines of Fig 2 and the four U87 conditions of Fig 5A,B), the highest rank for the driver kinases was either equivalent or better for uncorrected INKA score calculations. Therefore, we decided not to incorporate the iBAQ approach into INKA; see also Appendix Fig S13

-The kinase activation loop phosphopeptides were already used in the calculation of the first part of their score. It would seem wrong to use it twice. It would be instructive to see whether the INKA score would drastically change would one leave out the second part of their kinase-centric analysis.

Answer

The reviewer is formally right, yet by having it as a separate component in the score we can give extra weight to this important regulatory part of the kinase. The key objective of INKA is to quantify kinase activity from phosphoproteomics data. Therefore, we give a higher weight to the

phosphopeptides that originate from the activation loop. Phosphorylation of this part of the kinase sequence/structure is positively correlated with kinase catalytic activity. We added a sentence to clarify this.

Added text (pg 6 ln 86-89)

Although all kinase-derived phosphopeptides are already used in the first analysis above, here only phosphorylation of a kinase domain essential for kinase catalytic activity is considered for scoring, effectively doubling its contribution to the INKA score as a weighing measure.

- Moreover, their score will scale with measurement depth. This means that if they happen to measure a tumour sample on one day on a clean instrument and get many phosphopeptides for say ABL1 and its substrates (high INKA score), the same run on a poorly performing instrument will result in a lower INKA score. Dealing with such technical things may be easy in a well controlled lab environment but as the patient data in Figure 4 shows, the range of INKA scores can be quite different from sample to sample. Related, the authors should discuss how INKA scores are (not) comparable between laboratories, which is suboptimal for clinical decision making.

Answer

Indeed INKA scores are a function of measurement depth in a phosphoproteomics experiment. We perform our phosphoproteomics experiments using strict protocols for both sample preparation and MS data acquisition. However, key in the INKA analysis is the ranking procedure, which does not depend on the absolute INKA score values. An INKA score of a kinase is only relevant in the context of other kinases and their INKA scores. Indeed upon future implementation in clinical practice, standardization of sample prep and MS procedures is important to make INKA scores comparable between labs. To deal with varying INKA scores between experiments and labs, one can also normalize on the max INKA score. This simple procedure makes INKA scores comparable between experiments and possibly between labs. We actually applied this normalization in Fig 6.

We expanded the discussion with a sentence clarifying the above.

Added text (pg 19/20 ln 389-391)

Additionally, when analyzing INKA scores of different experiments or laboratories, INKA normalization on the maximum INKA score may standardize scores and allow comparison of data sets.

- All of their scores are based on spectral counts, which are - as they claim - less sensitive to outlier phosphopeptides with very high abundance. But maybe it's these outliers that are actually interesting! Related: because the data is available, the authors should check if INKA can be improved when using the peptide intensity provided by MaxQuant. Spectral counts are semi-quantitative at best and only work (well) for high spectral counts.

Answer

The INKA analysis using the intensity data for the cell lines in Fig 2 is now provided in Appendix Fig 12. Comparison of the analyses shows that count-based INKA outperforms intensity-based INKA in the task of ranking oncogenic driver kinases high in the 4 cancer cell lines in Fig 2. In 2/4 cases counts and intensities showed the same (top-3) driver kinase rank and in 2/4 cases count-based INKA ranked the driver kinase substantially higher than the intensity based analysis. Additionally, we plotted individual biological replicate analyses that show highly similar ranking and INKA scores across replicates.

Added text (Pg40 ln 851-857)

INKA: spectral counts versus Intensity

Although INKA analysis can be performed with intensity-based quantification, we favor spectral count-based quantification as it is less sensitive to peptides with outlier intensities and is more robust for the analysis of aggregated data for multiple peptides, some of which may exhibit dominantly high intensities. For Q Exactive data, spectral counting outperformed intensity-based quantification for INKA-based kinase ranking of known drivers (Appendix Fig S12), yet for the low level LTQ-FTMS data the intensity data worked better (Fig 5E,F).

Obviously, using LC-based quantification only works if the LC and MS parameters have been matched such that enough data points were collected across the LC peak. I could not get to this

information.

Answer

LC-MS based label-free quantification was based on the MS1 extracted ion chromatograms as implemented in MaxQuant. The median number of datapoints for intensity-based quantification is 31 based on the MaxQuant MS scans.txt file for the experiments shown in Fig 2. Conservatively, a minimum of 10 data-points over the eluting peak is required for correct quantification. We added this information in the Materials and methods section

Added text (Pg31 ln 650-651)

For the data in Fig 2, the average number of datapoints over the eluting peak is 31.

Related, the paper is in need for discussing the shortcomings of SC. This particularly shows in the analysis of the patient data in Figure 4 where INKA scores are quite low and vary between samples quite a lot. I understand that INKA provides a ranked list of kinases to consider but it is a stretch to interpret the EGFR finding in Figure 4C and 4D given the low INKA scores (I guess few p-peptides).

Answer

*Indeed the reviewer is right, at low level the intensity data may perform better, as we saw for the differential analysis example shown in figures 5E and 5F, where we applied INKA to intensity data because of older FT-MS data with a lower number of PSMs compared to orbitrap data. Having said that, spectral counting is a good quantification method for aggregated data, such as INKA, combining many observations without a single phosphopeptide dominating the final score. When only a limited number of spectra is identified for a kinase or substrate, then spectral counting will only approximate the correct quantitative value and intensity-based quantification may be better. The key ingredient for spectral counting to work is data aggregation: count values >10 are as good for quantitation as intensity data. This aggregation is exactly what INKA does. For our low input patient samples (Fig 5 C and D) the top 10 INKA scores are >10 counts and usable. Below 10 the correlation with intensity values decreases. **In Pg40 ln 856-862** counts vs intensity is discussed*

Added text (Pg40 ln 856-862)

Although INKA analysis can be performed with intensity-based quantification, we favor spectral count-based quantification as it is less sensitive to peptides with outlier intensities and is more robust for the analysis of aggregated data for multiple peptides, some of which may exhibit dominantly high intensities. For Q Exactive data, spectral counting outperformed intensity-based quantification for INKA-based kinase ranking of known drivers (Appendix Fig S_12), yet for the low level LTQ-FTMS data the intensity data worked better (Fig 5E,F)].

Looking at the kinases on the list of which quite a few are ranked higher than EGFR, a SRC family inhibitor such as Dasatinib would have made more sense than Erlotinib.

Answer

The patient samples came from a phase I clinical study with the purpose to study intra-tumor drug concentrations and they were not assigned to erlotinib treatment based on their molecular profile. We added text to clarify this in the results section

Added text (pg 13 ln 238-246)

*Biopsies were collected both before and after two weeks of erlotinib treatment to study intra-tumor drug concentrations within the framework of a phase I clinical study (standard dose, trial NCT01636908; Labots et al., submitted for publication). Patients were not assigned to erlotinib treatment based on molecular profiling. Nonetheless, the on-treatment biopsy from a patient with advanced head and neck squamous cell carcinoma showed a reduced INKA score and rank for EGFR as well as cell cycle-associated kinases (Fig 5C). Interestingly, in a pancreatic cancer patient, no residual EGFR activity could be inferred by INKA in a tumor biopsy after erlotinib treatment (Fig 5D). **The limited patient material that was available precluded replicate analysis so results reported here are preliminary.***

- In general, the quality of the data is sometimes hard to judge. Again, as interesting as **Figure 4** is, we do not know how significant the observed changes are. In the EGFR overexpression system, the

data is probably fairly robust (judging from high INKA scores implying high spectrum counts) but this is much less clear in the patient data. This is particularly important as the drug treatment of the cell lines always led to reduced (or unchanged) INKA scores but in the patient data there are cases with increased INKA scores. These are hard to interpret in the absence of replicates/error bars/confidence levels. For the same reason, it is hard to interpret the extent of INKA reduction for EGFR because it is unclear how many spectral counts underpin the INKA calculation

Answer

The referee appears to refer to Figure 5.

The oncogenic driver cell line data were acquired as bio-replicates and the individual bio-replicate data shows that the INKA analysis is robust indeed. We now show the analysis of the individual replicates in Appendix Fig S12. Yet the limited material for the patient data did not allow for replicate analysis so these results are preliminary. We have added a sentence to explain this on Pg 13 ln 238-246. (see answer above)

- They calculate p-values for their scores based on empirical null distributions from permutations. And then they claim that the correlation of low p-values with high INKA-scores "underscores the relevance of the INKA score". However, this is a self-fulfilling prophecy. The higher any of the individual scores contributing to INKA are (kinase-centric, activation loop, PSP, NWK), the less likely it will be to replace it with an even higher score from the permutations. This has nothing to do with the relevance of the INKA score.

Answer

We suppose the word relevance is not correctly used here in connection with the p-value calculation. We have changed the text to be more neutral by deleting the second part of the sentence (...underscores the relevance of the INKA score)

Added text (Pg 10 ln 165)

Higher INKA scores clearly correlate with lower p-values (Appendix Figure S6).

- They then go ahead and say the INKA score can be used in differential analyses and that it predicts drug sensitivity. However, they fail to actually correlate the INKA score to the GDSC data or do something like ElasticNet regression. Would an approach like the ElasticNet select e.g. ABL1 based on INKA score when looking at, say, imatinib? I appreciate that the number of cell lines available in their data set may preclude such an analysis. However, at least for some of the cell lines included in the GDSC panel, there is public p-proteome data.

Answer

The reviewer raises an important point. We have now added an analysis of a panel of cancer cell lines for which drug data is available. In the new results section 'Comparing INKA to its components, to KARP, and correlation with drug efficacy' we describe the analysis to compare INKA, its components and KARP (wilkes et al MCP 2017), a single-sample substrate-based kinase activity inference method. We introduce the kinase impact score and apply this algorithm to the kinase ranking methods above. The kinase impact score correlates kinase activity ranked lists with cell line IC50 values of drugs in GDSC. We used publicly available pTyr IP data as well as TiO₂ global phosphoproteomics data, in addition to cell lines described in this manuscript, in two separate analyses with in total 20 distinct cell lines and a statistical analysis of significance. The kinase impact score is globally higher for INKA than for KARP, or for the four INKA components, indicating that INKA is superior in ranking kinase activities in the context of drug efficacy (Fig 6).

We added the new Fig 6 where the results of this analysis are shown, as well as a results section (Pg 14-15 ln 270-298), and expanded the materials and methods section (Pg 38-40 ln 810-849) accordingly.

- The authors claim that it is a good idea to calculate the INKA score only based on kinase phosphosites when there are no substrates known, but don't calculate INKA score when a kinase was not detected but their substrates were present, which makes sense. They say: "For kinases inferred through PhosphoSitePlus/NetworKIN but not observed by MS, the reciprocal analysis is not

performed, as kinases display overlapping substrate specificities precluding unequivocal assignment of a substrate to a specific kinase." However, they then go ahead and interpret their data on SK-MEL-28 (BRAfV600E) in the light of downstream substrates of BRAf and claim that high INKA scores of downstream kinases means the upstream kinase must have been active. Maybe they should do regular (e.g. IMAC or TiO₂) phosphopeptide enrichment and not pY-IPs and see whether BRAf ends up with the top INKA score.

Answer

The reviewer raised a relevant point. We have performed the TiO_x experiment on the SK-MEL-28 cell line, yet we did not identify BRAf at all, indicating that it may be below our detection threshold.

- When looking at HCC827-ER3, they say that these cells are highly sensitive to EGFR inhibitors and refer to Supplementary Table 4. There, -1.99 and even -1.17 (z-score) is apparently highly sensitive. Later, they focus on ALK and alectinib and say -1.92 (again z-score) is not sensitive. Since the z-score makes sensitivities somewhat comparable between drugs, they interpret the same data differently depending on their expectations.

Answer

Based on the comments given by the reviewer, we realize that including the Z-score in the tables raises confusion. The claims made in the paper are made based on the IC₅₀ value rather than the Z-score. The latter reflects the relative sensitivity of a cell line to a given drug compared to all other cell lines in the database. The Z-score indeed allows for comparison between cell lines, however our claims with regard to sensitivity of the cell lines are based on the specific IC₅₀ value for a given cell line. In general, an IC₅₀ in the low nano-molar range is regarded as sensitive. With regard to the Z-score, we decided to remove these data from the table as they do not add to the point we want to make and only result in confusion.

- Figure 6 and Supplementary Figure 9 are identical. I therefore cannot judge Supplementary Figure 9. However, in Figure 6D, they claim that treating PDX with a selection of drugs targeting kinases with high INKA scores highlights the INKA score as a potential tool for personalized medicine.

Answer

We apologize for the mistake. We now provide the correct figure.

- However, they fail to check whether these drugs would have also killed PDX in which other kinases scored high using INKA! Maybe these compounds just generally work well in their PDX model.

Answer

We repeated the IC₅₀ determination in organoids for afatinib and BMS 754807, and also included imatinib for PDX0177 and PDX0254. Imatinib targets (BCR-ABL and SRC-family members) rank low by INKA in these PDX models and their corresponding organoids. IC₅₀ values for CRC0177 and CRC0254 were 4 and 6 μ M, respectively, indicating low efficacy of imatinib in these organoids. Fig 7 has been updated with these drug-response data and in appendix figure S11 the IC₅₀ determination is shown.

Added text (17 ln326-331)

To explore whether a kinase with a low INKA score does not show a response to the corresponding drug, we selected ABL that ranked low in both PDXs and organoids of CRC0177 and CRC0254. Indeed, organoid treatment with the ABL inhibitor imatinib yielded negligible inhibition (IC₅₀ imatinib = 4 or 6 μ M for CRC0177 and CRC0254, respectively) (Fig. 7D) while the positive control (CML cell line K562) worked (supplementary figure S11E), underscoring the value of INKA ranking for drug response prediction.

- There is no direct comparison to e.g. KSEA.

Answer

KSEA is not a single-sample workflow and therefore we did not compare INKA to KSEA. However, now we include a comparison of INKA to KARP. To our knowledge, KARP is the only other single-sample workflow and makes use of substrate data to infer kinase activity, in a way that is

comparable to our “PSP” arm, yet with some differences in the analysis. To this end, we implemented and ran KARP on the 4 cancer cell lines of Figure 2. The comparison of INKA vs KARP (Appendix Fig S 7) shows the superiority of INKA over KARP for oncogene-driven cell lines in (high) ranking the driver kinase(s). The text has been modified

Added text (pg 10 ln 165-170)

Higher INKA scores clearly correlate with lower p-values (Appendix Figure S6). The INKA score was compared to KARP (Wilkes et al, 2017), another kinase activity ranking tool that can be used on single samples. KARP kinase activity ranking is based on substrate phosphorylation analysis in combination with kinase-substrate relations. For the four oncogene-driven cell lines, INKA outperformed KARP in assigning high ranks to the known drivers (Appendix Fig S7).

- I could not use their website, since it requires a login.

Answer

The login was provided in the cover letter and is “Username: inkareviewer
Password: oncoproteomics”

- Use of the word phosphokinases in the title is weird.

Answer

We changed the word phosphokinase to kinase

- Manuscript suffers from poor use of the English language in some paragraphs.

Answer

We corrected the English.

4th Editorial Decision

25th February 2019

Thank you for sending us your revised manuscript. We have now heard back from the two referees who agreed to evaluate your study. As you will see below, the reviewers think that the study has improved as a result of the performed revisions. However, reviewer #2 raises some remaining concerns, which we would ask you to address in a revision. Reviewer #2 provides specific suggestions on what remains to be addressed.

REFEREE REPORTS

Reviewer #2:

The authors provide a revised version that addresses in part our comments as well as those of other reviewers.

We feel however that the response and revisions fall short in addressing several points, as we outline below:

Major points:

1# Authors clearly state that their method is only suited for label-free phosphoproteomics. This is a major limitation, since as the authors state in their response to point 4 to Rev 1: 'Only very few label-free phosphoproteomics datasets are available in the public domain and they are limited in size' this limits significantly the applicability and thus significance of INKA, given the limited number of these data sets. In addition, since the method works on spectral counts (and there might be good reasons for this), this further limits to which data sets it can be applied. Couldn't authors modify the methods to use broader types of data types? Or at least explain how this could be attempted?

2# Response to Major Point 1 of Rev 2:

However, this dataset as such cannot be used as input for the INKA pipeline, as that was not

designed to perform group comparisons (fold-change values).

To us is not clear why authors can not use the data in the benchmark. They could for example rank single samples and then do ranking comparisons between groups.

3#

Providing access to tool: this is only addressed - Authors state: that the R script is available for download at www.inkascore.org But we could not find it. The code in fact should be provided with the paper as supplementary material, and ideally in a public repository such as CRAN.

4#

Rev 2 Point 3 on the motivation of the value on single sample. Authors simply quote reviewer 3. We had hoped an actual explanation.

In fact, we do not agree with their statement about single sample analysis. Most enrichment based analysis can be adapted to single samples, even if explicitly this was only done before by KARP for this specific topic. In addition, group comparisons could be used downstream of their method so that it can be used as classic methods.

Furthermore, we usually don't stop at single samples because to treat a disease we can not just target what is the most active in a disease. We need to compare that with a healthy reference so that you can differentiate between what is a general biological mechanism and what is driving a disease.

5#

Comment on rev. 3: The kinase activation loop phosphopeptides were already used in the calculation of the first part of their score. ...

Authors should have done as reviewer #3 suggest and run the analysis using it only once.

Minor points:

1#

Comment on measurement depth of Rev 3:
here authors could/should have done e.g.a subsampling analysis

2# Comment Rev 3 on 'relevance' of score, the authors could have elaborated and further clarified this point, rather than just removing the word 'relevance'

Reviewer #3:

The authors have done a good job to respond to the concerns raised by the reviewers. They did new experiments and performed new data analysis and made many changes to the original manuscript. They do not agree with all the points raised nor would I agree with all the answers but overall, I think the revision is fine. As I wrote in my first report, methods for the interpretation of single sample datasets are deerly needed and INKA, albeit not perfect, makes an important step in this direction.

2nd Revision - authors' response

15th March 2019

Point-by-Point Response

for MSB-18-8250RRR, "INKA, an integrative data analysis pipeline for phosphoproteomic inference of active kinases"

Again, we thank the reviewers for their time to read our manuscript and their valuable comments. Below we provide a detailed reply to the remaining points raised by the reviewers.

Reviewer #2:

The authors provide a revised version that addresses in part our comments as well as those of other reviewers. We feel however that the response and revisions fall short in addressing several points, as we outline below:

Major points:

1# Authors clearly state that their method is only suited for label-free phosphoproteomics. This is a major limitation, since as the authors state in their response to point 4 to Rev 1: 'Only very few label-free phosphoproteomics datasets are available in the public domain and they are limited in size' this limits significantly the applicability and thus significance of INKA, given the limited number of these data sets. In addition, since the method works on spectral counts (and there might be good reasons for this), this further limits to which data sets it can be applied. Couldn't authors modify the methods to use broader types of data types? Or at least explain how this could be attempted?

ANSWER:

We would like to clarify that our method is not only suited for label-free phosphoproteomics data, but can be adapted to analyze labeled phosphoproteomics data as well. To demonstrate this, we have adapted INKA to 11-plex TMT (isobaric labeling) data, and performed a re-analysis of deep phosphoproteome data of the Olsen team (Emdal et al., *Sci Signal.* 2018 Nov 20;11, 557) who employed TMT labeling of cancer cell lines to explore the effect of ALK inhibition.

In section "*Testing the INKA approach in differential settings*" (lines 266-273) and the corresponding Materials and Methods/INKA analysis section ("*INKA analysis of 11-plex isobaric TMT data*", lines 882-888), we describe the INKA analysis of this TMT experiment, in which we find the whole ALK pathway down-modulated in response to kinase inhibitor treatment or siRNA down-modulation. We added an extra supplementary figure (Appendix Figure S14), and included the reference to the Emdal *et al.* paper.

Furthermore, though we favor application of the INKA method to spectral counts as detailed in section "*INKA: spectral counts versus Intensity*" (lines 859-865), we extended the method to include analyses employing intensity data, as we report in Fig 5E,F (section "*Testing the INKA approach in differential settings*", lines 258-265) for the re-analysis of the DNA damage experiment, and in Appendix Fig S12 (section "*INKA: spectral counts versus Intensity*", line 864) for the cancer cell use cases.

2# Response to Major Point 1 of Rev 2:

However, this dataset as such cannot be used as input for the INKA pipeline, as that was not designed to perform group comparisons (fold-change values). To us is not clear why authors can not use the data in the benchmark. They could for example rank single samples and then do ranking comparisons between groups.

ANSWER:

We recognize the value of a large phosphorylated peptide resource and proceeded to download and install the database tables. Unfortunately, we were only able to find relative quantitation measures associated with all peptides in the form of log₂-transformed values. Lacking an absolute scale, we were not able to transform these values into absolute quantities, which are the type of quantities INKA was devised for.

3# Providing access to tool: this is only addressed - Authors state: that the R script is available for download at www.inkascore.org But we could not find it. The code in fact should be provided with the paper as supplementary material, and ideally in a public repository such as CRAN.

ANSWER:

In our previous cover letter we mentioned that the code can be downloaded as a zip file from our website (http://www.inkascore.org/inka_code.zip). This is where the R script can be downloaded.

4# Rev 2 Point 3 on the motivation of the value on single sample. Authors simply quote reviewer 3. We had hoped an actual explanation. In fact, we do not agree with their statement about single sample analysis. Most enrichment based analysis can be adapted to single samples, even if explicitly this was only done before by KARP for this specific topic. In addition, group comparisons could be used downstream of their method so that it can be used as classic methods.

Furthermore, we usually don't stop at single samples because to treat a disease we can not just target what is the most active in a disease. We need to compare that with a healthy reference so that you can differentiate between what is a general biological mechanism and what is driving a disease.

ANSWER:

In the setting of clinical oncology and precision medicine, single-sample analysis is extremely relevant, as the oncologist needs to make treatment decisions based on a molecular profile of an individual patient. Of course, in the preclinical setting, and in the setting of biomarker discovery and test validation, group-based comparisons are very relevant as well. Many tools exist for differential group-based analysis. That is why we focused here on a tool that enables single-sample analysis. And, as we show, this tool can be harnessed for differential analysis as well (section “*Testing the INKA approach in differential settings*” : Figure 5 (lines 224-265) and Appendix Fig S14, lines 266-273).

5# Comment on rev. 3: The kinase activation loop phosphopeptides were already used in the calculation of the first part of their score. ... Authors should have done as reviewer #3 suggest and run the analysis using it only once.

Minor points:

1# Comment on measurement depth of Rev 3: here authors could/should have done e.g.a subsampling analysis

2# Comment Rev 3 on 'relevance' of score, the authors could have elaborated and further clarified this point, rather than just removing the word 'relevance'

ANSWER:

For the answers to the questions of reviewer 3 that are repeated here I would like to refer the reviewer to the answers in our previous rebuttal.

Reviewer #3:

The authors have done a good job to respond to the concerns raised by the reviewers. They did new experiments and performed new data analysis and made many changes to the original manuscript. They do not agree with all the points raised nor would I agree with all the answers but overall, I think the revision is fine. As I wrote in my first report, methods for the interpretation of single sample datasets are deeply needed and INKA, albeit not perfect, makes an important step in this direction.

ANSWER:

We are happy to read this feedback.

Accepted

20th March 2019

Thank you again for sending us your revised manuscript. We are now satisfied with the modifications made and I am pleased to inform you that your paper has been accepted for publication.

Corresponding Author Name: Connie R. Jimenez

Manuscript Number: MSB-18-8250RR